# Streamlining Prediction in Bayesian Deep Learning

**Rui Li**     **Marcus Klasson**     **Arno Solin**     **Martin Trapp**
Department of Computer Science, Aalto University, Finland
`{firstname.lastname}@aalto.fi`

## Abstract

The rising interest in Bayesian deep learning (BDL) has led to a plethora of methods for estimating the posterior distribution. However, efficient computation of inferences, such as predictions, has been largely overlooked with Monte Carlo integration remaining the standard. In this work we examine streamlining prediction in BDL through a single forward pass without sampling. For this, we use local linearisation of activation functions and local Gaussian approximations at linear layers. Thus allowing us to analytically compute an approximation of the posterior predictive distribution. We showcase our approach for both MLP and transformers, such as ViT and GPT-2, and assess its performance on regression and classification tasks.

Open-source library: `https://github.com/AaltoML/SUQ`.

## 1 Introduction

Recent progress and adoption of deep learning models have led to a sharp increase in interest of improving their reliability and robustness. In applications such as aided medical diagnosis (Begoli et al., 2019), autonomous driving (Michelmore et al., 2020), or supporting scientific discovery (Psaros et al., 2023), providing reliable and robust predictions as well as identifying failure modes is vital. A principled approach to address these challenges is the use of Bayesian deep learning (BDL, Wilson & Izmailov, 2020; Papamarkou et al., 2024) which promises a *plug & play* framework for uncertainty quantification. However, while *plugging* the Bayesian approach into deep learning is relatively straightforward (Blundell et al., 2015; Gal & Ghahramani, 2016; Wu et al., 2019), the *play* part is typically severely hampered by computational and practical challenges (Wenzel et al., 2020; Foong et al., 2020; Gelberg et al., 2024; Coker et al., 2022; Kristiadi et al., 2023).

The key challenges associated with BDL can roughly be divided into three parts: *(i)* defining a meaningful prior, *(ii)* estimating the posterior distribution, and *(iii)* performing inferences of interest, *e.g.*, making predictions for unseen data, detecting out-of-distribution settings, or analysing model sensitivities. While constructing a meaningful prior is an important research direction (Nalisnick, 2018; Meronen et al., 2021; Fortuin et al., 2021; Tran et al., 2022), it has been argued that the

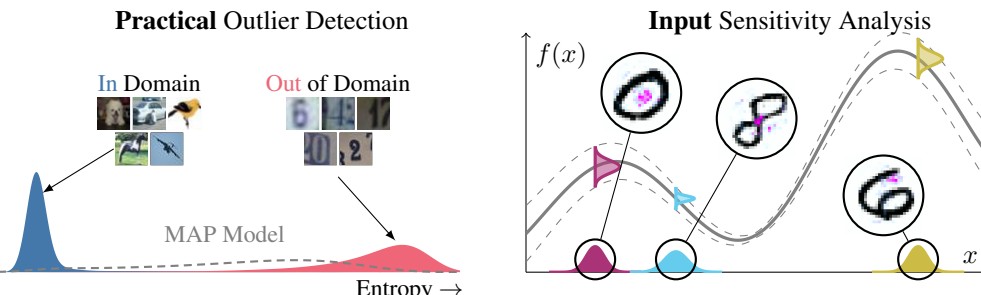

Figure 1: Our streamlined approach allows for *practical* outlier detection and sensitivity analysis. Locally linearising the network function with local Gaussian approximations enables many relevant inference tasks to be solved analytically, helping render BDL a practical tool for downstream tasks.

differentiating aspect of Bayesian deep learning is marginalisation (Wilson & Izmailov, 2020; Wilson, 2020) rather than the prior itself. Hence, estimating the posterior distribution has seen significant progress in recent years (Blundell et al., 2015; Maddox et al., 2019; Daxberger et al., 2021a) with a particular focus on post-hoc approximations (Kristiadi et al., 2020; Daxberger et al., 2021b). However, while these approaches have shown promise in making BDL useful for real-world applications, they tackle only part of the computational and practical challenges associated with BDL.

In this work, we focus on streamlining prediction in BDL for downstream tasks by providing a straightforward and effective method to compute inferences of interest, *cf.*, Fig. 1. For this, we make the neural network locally linear with respect to the inputs. Thus, inferences, such as computing predictions, admit a closed-form solution and can be estimated efficiently. In particular, we propose using local linearisation of non-linear activation functions at every layer of the network and local Gaussian approximations at linear layers. Empirically, we find that local linearisation combined with Gaussian approximation of Bayesian neural networks provides accurate predictions, with useful predictive uncertainties, while being conceptually simple. Moreover, complex inference tasks w.r.t. the inputs, such as analysing model sensitivities to input perturbations, can be computed efficiently. Thus allowing us to truly account for all sources of uncertainties.

**Contributions:** *(i)* We propose layer-wise local linearisation and local Gaussian approximations of neural networks to streamline BDL for downstream tasks (Sec. 3). *(ii)* We discuss how to handle different covariance structures and architecture choices (Sec. 3.2 & Sec. 3.3). *(iii)* Finally, we present an empirical assessment of our approach on regression and classification tasks, and showcase its utility for uncertainty quantification, out-of-domain detection, and sensitivity analysis (Sec. 4).

## 2 RELATED WORK

To estimate the posterior in BDL, variational inference (VI, Blei et al., 2017; Zhang et al., 2018) utilises a variational approximation to the true posterior distribution and minimises a divergence measure between both distributions. A typical choice for the variational family is a factorised Gaussian distribution, chosen for computational reasons. Early works on mean-field VI (MFVI) and related approaches require modifications of the model structure (Blundell et al., 2015) to perform a reparametrisation of the variational distribution. Recent work by Shen et al. (2024) developed an optimiser to ease the use of MFVI, and has shown good performance on large-scale models such as ResNets (He et al., 2016) and GPT-2 (Radford et al., 2019). However, VI-based methods typically require Monte Carlo estimation to perform inferences, which can be problematic in practice due to additional computational overhead.

A recent trend in BDL are post-hoc methods, such as the Laplace approximation (LA, MacKay, 1992a), which can be applied directly on the trained model without modification (Kristiadi et al., 2020; Daxberger et al., 2021a). Daxberger et al. (2021b) extended the applicability of LAs by showing that treating a subset of parameters Bayesian can still give good predictive uncertainties. Moreover, Immer et al. (2021b) proposed the linearised LA by performing a global linearisation, which is principled under the Generalised Gauss–Newton approximation to the Hessian, and has shown promise in providing useful predictive uncertainties. Recent works applied post-hoc methods in various applications, such as large language models (Yang et al., 2024; Kampen et al., 2024), vision-language models (Baumann et al., 2024), dynamic neural networks (Meronen et al., 2024), and sequential learning (Scannell et al., 2024).

In addition, various tailored ensemble-based methods for BDL have been proposed, such as Monte Carlo dropout (Gal & Ghahramani, 2016), deep ensembles (Lakshminarayanan et al., 2017), and stochastic weight averaging-Gaussian (Maddox et al., 2019). While some works on deep ensembles enable estimating the predictive distribution in a single forward pass (Eschenhagen et al., 2021; Havasi et al., 2021), most methods typically require multiple forward passes to estimate the predictive distribution and do not explicate an approximation to the posterior distribution.

More recently, there has been a trend in exploring deterministic computations in BDL to avoid the need for sampling (Goulet et al., 2021; Giordano et al., 2024; Burroni et al., 2024). In particular, Wu et al. (2019) derived an analytical training objective for VI by using moment-matching at each layer of the network. However, the solutions to the moment-matching have to be derived manually for each type of activation function, making it impractical in practice. More recently, Goulet et al.

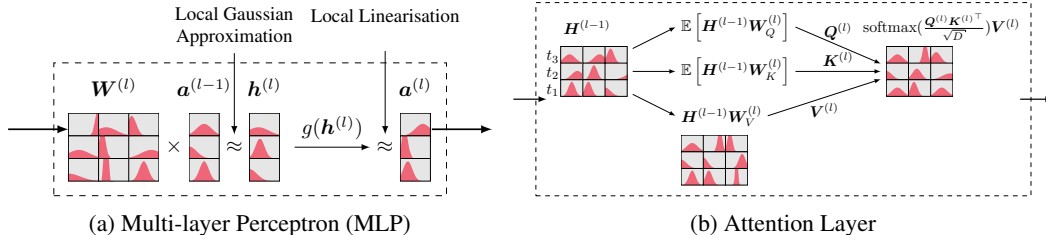

(a) Multi-layer Perceptron (MLP)        (b) Attention Layer

Figure 2: Illustration of our approach for different network architectures. In MLPs, we can directly apply local Gaussian approximations and local linearisation of each layer. The distribution over activations is then propagated to the next layer. In attention layers, we treat the query $Q$ and key $K$ deterministically and only treat the value $V$ as a random quantity, resulting in a straightforward propagation path. The resulting distribution is then propagated to the subsequent MLP layer.

(2021) proposed local linearisation of the network to perform message passing on the network under a mean-field assumption. Moreover, Petersen et al. (2024) used a local linearisation of the network to propagate aleatoric uncertainties over the input through a deterministic network. In addition, Dhawan et al. (2023) investigated local linearisations of activation functions to estimate the function space distance of two neural networks, for example, relevant in continual learning settings. In contrast, our work disentangles the approximation of the posterior distribution and the computation of inferences w.r.t. the posterior distribution. Hence, it provides a streamlined framework to propagate all forms of uncertainties through Bayesian neural networks.

## 3 METHOD

In Bayesian deep learning (BDL), predicting the output $y$ (*e.g.*, class label, regression value) for an input $x \in \mathcal{X}$ is performed by *marginalising* out the model parameters $\theta$ of the neural network $f_\theta(\cdot)$ instead of trusting a single point estimate, *i.e.*,

$$p(y \,|\, x, \mathcal{D}) = \int_\theta p(y \,|\, f_\theta(x)) \, p(\theta \,|\, \mathcal{D}) \, \mathrm{d}\theta, \tag{1}$$

where $\mathcal{D} = \{(x_n, y_n)\}_{n=1}^N$ denotes the training data and the posterior distribution $p(\theta \,|\, \mathcal{D}) = \frac{p(\theta, \mathcal{D})}{p(\mathcal{D})}$ is given by Bayes' rule. However, for most neural networks, integrating over the high-dimensional parameter space is intractable, necessitating the use of approximations to compute the posterior distribution $p(\theta \,|\, \mathcal{D})$ and the posterior predictive distribution $p(y \,|\, x, \mathcal{D})$.

Recently, much progress has been made in efficiently approximating the posterior distribution for BDL, including scaling mean-field variational inference (Shen et al., 2024) to large-scale models and performing post-hoc estimation using the Laplace approximations (Daxberger et al., 2021a). A common thread is using a tractable distribution $q$ to approximate the posterior distribution $q(\theta) \approx p(\theta \,|\, \mathcal{D})$, commonly chosen as a Gaussian distribution. Consequently, the posterior predictive distribution is typically approximated using Monte Carlo integration, *i.e.*, by sampling from $q$, to estimate the integral in Eq. (1), with the exception of the linearised Laplace approximation (Immer et al., 2021b). However, while using a Gaussian approximation facilitates efficient computation of the approximate posterior distribution, sampling from the high-dimensional Gaussian approximation can be challenging (Vono et al., 2022) and result in high computational overhead.

We will now shift our focus on estimating integrals of the form of Eq. (1) and assume that an approximation to the posterior distribution $q(\theta)$ is given. Further, we will assume that $q$ is in the family of stable distributions. Note that a linear combination of two independent random variables following a stable distribution has the same distribution as the distribution of the individual random variables. The Gaussian distribution is a typical example of a stable distribution. Marginalisation tasks such as in Eq. (1) appear in many scenarios, *e.g.*, active learning (MacKay, 1992a; Gal et al., 2017; Smith et al., 2023), model selection (Immer et al., 2021a; MacKay, 1996), or outlier detection (Wilson & Izmailov, 2020), and pose a reappearing challenge in downstream applications of BDL.

## 3.1 STREAMLINING COMPUTATIONS WITH LOCAL APPROXIMATIONS

Let the weights and biases of the $m^{\text{th}}$ linear layer of the network $f$ be denoted as $\boldsymbol{W}^{(m)} \in \mathbb{R}^{D_{\text{out}} \times D_{\text{in}}}$ and $\boldsymbol{b}^{(m)} \in \mathbb{R}^{D_{\text{out}}}$, respectively. Then the pre-activation $\boldsymbol{h}^{(m)}$ is given as $\boldsymbol{h}^{(m)} = \boldsymbol{W}^{(m)} \boldsymbol{a}^{(m-1)} + \boldsymbol{b}^{(m)}$, where $\boldsymbol{a}^{(m-1)} \in \mathbb{R}^{D_{\text{in}}}$ is the activation of the previous layer. In case $m = 1$, then $\boldsymbol{a}^{(0)}$ corresponds to the input $\boldsymbol{x}$. We further denote the $k^{\text{th}}$ element of $\boldsymbol{h}^{(m)}$ as $h_k^{(m)} = \sum_{i=1}^{D_{\text{in}}} W_{ki}^{(m)} a_i^{(m-1)} + b_k^{(m)}$ and drop the superscript if it is clear from the context.

Given an approximate posterior distribution $q(\boldsymbol{\theta})$ with $\boldsymbol{\theta} = \{\boldsymbol{W}^{(m)}, \boldsymbol{b}^{(m)}\}_{m=1}^{M}$, we aim to compute the probability distribution of the activation $\boldsymbol{a}^{(m)}$ of each layer $m$. For this, we need to estimate the distribution of the pre-activation $\boldsymbol{h}^{(m)}$ and then compute an approximation to the activation $\boldsymbol{a}^{(m)}$ after the application of a non-linear activation function $g(\cdot)$.

**Approximating the pre-activation distribution** In case the activation $\boldsymbol{a}^{(m-1)}$ is deterministically given, *i.e.*, for the input layer, we can compute the distribution over pre-activations analytically as a consequence of the stability of stable distributions under linear transformations (Petersen et al., 2024). However, for hidden layers, the distribution over pre-activations is generally not of the same family as the posterior distribution (Wolinski & Arbel, 2022). Nevertheless, we will apply a local Gaussian approximation to the pre-activation at every hidden layer. Specifically, we make the assumption:

**Assumption 3.1.** *Assume that the activations of the previous layer $a_i^{(m-1)}$ and parameters of the $m^{th}$ layer are independent.*

Then followed by a Gaussian approximation of $a_i^{(m-1)} W_{ki}^{(m)}$ for each $i$ and each $k$, the mean of the pre-activation $\boldsymbol{h}^{(m)}$ is given as:

$$\mathbb{E}\left[\boldsymbol{h}^{(m)}\right] = \mathbb{E}\left[\boldsymbol{W}^{(m)}\right] \mathbb{E}\left[\boldsymbol{a}^{(m-1)}\right] + \mathbb{E}\left[\boldsymbol{b}^{(m)}\right], \tag{2}$$

and the covariance between the $k^{\text{th}}$ and the $j^{\text{th}}$ hidden unit is computed as:

$$\begin{aligned}
\mathbb{C}\text{ov}\left[h_k^{(m)}, h_l^{(m)}\right] = &\sum_{1 \leq i,j \leq D_{\text{in}}} \mathbb{C}\text{ov}\left[a_i^{(m-1)} W_{ki}^{(m)}, a_j^{(m-1)} W_{lj}^{(m)}\right] + \mathbb{C}\text{ov}\left[b_k^{(m)}, b_l^{(m)}\right] \\
&+ \sum_{1 \leq i \leq D_{\text{in}}} \mathbb{E}\left[a_i^{(m-1)}\right] \left(\mathbb{C}\text{ov}\left[W_{ki}^{(m)}, b_l^{(m)}\right] + \mathbb{C}\text{ov}\left[W_{li}^{(m)}, b_k^{(m)}\right]\right),
\end{aligned} \tag{3}$$

where

$$\begin{aligned}
\mathbb{C}\text{ov}\left[a_i^{(m-1)} W_{ki}^{(m)}, a_j^{(m-1)} W_{lj}^{(m)}\right] = &\mathbb{E}\left[a_i^{(m-1)}\right] \mathbb{E}\left[a_j^{(m-1)}\right] \mathbb{C}\text{ov}\left[W_{ki}^{(m)}, W_{lj}^{(m)}\right] \\
&+ \mathbb{E}\left[W_{ki}^{(m)}\right] \mathbb{E}\left[W_{lj}^{(m)}\right] \mathbb{C}\text{ov}\left[a_i^{(m-1)}, a_j^{(m-1)}\right] \\
&+ \mathbb{C}\text{ov}\left[a_i^{(m-1)}, a_j^{(m-1)}\right] \mathbb{C}\text{ov}\left[W_{ki}^{(m)}, W_{lj}^{(m)}\right].
\end{aligned} \tag{4}$$

A detailed derivation alongside an empirical evaluation of the approximation quality can be found in Apps. A.1 and A.2. Depending on the structure of the covariance matrix, we can further simplify the computation of the covariance matrix, which we will discuss in Sec. 3.3.

**Approximating the activation distribution** Let $g(\cdot)$ denote a non-linear activation function computing $\boldsymbol{a} = g(\boldsymbol{h})$ for a pre-activation $\boldsymbol{h}$. Inspired by the application of local linearisation in Bayesian filtering (*e.g.*, Särkkä & Svensson, 2023), we use a first-order Taylor expansion of $g(\cdot)$ at the mean of the pre-activation $\mathbb{E}[\boldsymbol{h}]$. Specifically, we approximate $g(\boldsymbol{h})$ using

$$g(\boldsymbol{h}) \approx g(\mathbb{E}[\boldsymbol{h}]) + \boldsymbol{J}_g|_{\boldsymbol{h}=\mathbb{E}[\boldsymbol{h}]}(\boldsymbol{h} - \mathbb{E}[\boldsymbol{h}]), \tag{5}$$

where $\boldsymbol{J}_g|_{\boldsymbol{h}=\mathbb{E}[\boldsymbol{h}]}$ is the Jacobian of $g(\cdot)$ at $\boldsymbol{h} = \mathbb{E}[\boldsymbol{h}]$. Then, as stable distributions are closed under linear transformations, the distribution of $\boldsymbol{a}$ can be computed analytically and is given as follows in case of a Gaussian distribution, *i.e.*,

$$\boldsymbol{a} \sim \mathcal{N}(g(\mathbb{E}[\boldsymbol{h}]), \boldsymbol{J}_g|_{\boldsymbol{h}=\mathbb{E}[\boldsymbol{h}]}^{\top} \boldsymbol{\Sigma}_{\boldsymbol{h}} \boldsymbol{J}_g|_{\boldsymbol{h}=\mathbb{E}[\boldsymbol{h}]}). \tag{6}$$

Note that the quality of the local linearisation will depend on the scale of the distribution over the input $\boldsymbol{h}$. For ReLU activation functions, Petersen et al. (2024) have shown that local linearisation

provides the optimal Gaussian approximation of a univariate Gaussian distribution in total variation. For classification tasks, we employ a probit approximation MacKay (1992b); Kristiadi et al. (2020).

**Intuition** One way to understand the resulting approximation is as a piecewise linear function (or multilinear function). Globally, the function will still be non-linear, but locally it will behave linearly. In contrast to the original model, which composes piecewise linear functions in the case of a ReLU network, our approximation composes linear functions locally. We obtain a piecewise linear function due to the local composition, which allows us to capture the non-linear nature of the model.

## 3.2 Architecture Choices

By combining local Gaussian approximations for linear layers and local linearisation for non-linear activation functions, we can analytically compute the distribution over activations at each layer in a single forward pass. In the case of a multi-layer perceptron (MLP) and common architecture choices, the described approach can be directly applied to each layer of the network. However, further considerations are needed to streamline the computation path for more complex architectures such as attention. Fig. 2 illustrates the computation path for MLPs and attention layers.

**Attention layers** Each block in a transformer (Vaswani et al., 2017) constitutes: multi-head attention, an MLP, layer normalisation, and a residual connection. For the MLP part, the propagation is the same as previously described. Further, layer normalisation is a linear transformations, and the resulting distribution can be obtained analytically. For residual connections by assuming independence, we could also obtain the resulting distribution analytically. Treating the multi-head attention block is more involved as the softmax activation function 'squashes' the distribution of the pre-activations. We describe our method below, and further details are in App. A.6.

Given an input $\boldsymbol{H} \in \mathbb{R}^{T \times D}$, where $T$ is the number of tokens in the input sequence and $D$ is the dimension of each token, denote the query, key and value matrices as $\boldsymbol{W}_Q \in \mathbb{R}^{D \times D}$, $\boldsymbol{W}_K \in \mathbb{R}^{D \times D}$, $\boldsymbol{W}_V \in \mathbb{R}^{D \times D}$, respectively. Further, we denote the key, query and value in an attention block as $\boldsymbol{Q} = \boldsymbol{H}\boldsymbol{W}_Q$, $\boldsymbol{K} = \boldsymbol{H}\boldsymbol{W}_K$, and $\boldsymbol{V} = \boldsymbol{H}\boldsymbol{W}_V$. Then the output of attention layer is given as follows $\text{Attention}(\boldsymbol{H}) = \text{Softmax}\left(\boldsymbol{Q}\boldsymbol{K}^\top/\sqrt{D}\right)\boldsymbol{V}$. For computational reasons, we will assume the input distribution to the multi-head attention block has a diagonal covariance structure. 'Pushing' random vectors over a softmax activation may require further approximations and will not result in an output with a distribution close to a Gaussian distribution. Hence, we treat the query and key matrices as deterministically given. A possible remedy is to leverage an approximation to the softmax function such as Lu et al. (2021). Consequently, the attention scores are given as:

$$\text{Attention}(\boldsymbol{H}) = \text{Softmax}\left(\frac{\mathbb{E}\left[\boldsymbol{H}\right]\mathbb{E}\left[\boldsymbol{W}_Q\right]\left(\mathbb{E}\left[\boldsymbol{H}\right]\mathbb{E}\left[\boldsymbol{W}_K\right]\right)^\top}{\sqrt{D}}\right)\boldsymbol{V}, \tag{7}$$

where $\boldsymbol{V}$ follows a stable distributions. Due to linearity, the resulting distribution can again be obtained analytically.

## 3.3 Covariance Structure

Computing the full covariance of the posterior is usually infeasible due to high computational and memory cost. We describe our methods for the two most common covariance approximations and will briefly discuss the computational cost in the case of a full and diagonal covariance structure.

**Full covariance** When the posterior has full covariance, for the $m^\text{th}$ linear layer the computational complexity for computing $\mathbb{C}\text{ov}[h_k, h_l]$ is $\mathcal{O}([D_\text{in}^{(m)}]^2)$. Consequently, computing the covariance of the activations for the $m^\text{th}$ layer adds to $\mathcal{O}([D_\text{out}^{(m)}]^2[D_\text{in}^{(m)}]^2)$. Computing the local linearisation for element-wise activation functions results in a complexity of $\mathcal{O}([D_\text{out}^{(l)}]^2)$. Hence, we obtain a total cost of $\mathcal{O}(\sum_{m=1}^{M}[D_\text{out}^{(m)}]^2[D_\text{in}^{(m)}]^2 + [D_\text{out}^{(m)}]^2)$ for a network with $M$ layers. As the computational cost is directly linked to the number of parameters and their correlation structure, a natural way to reduce the computational cost is to either exploit the structure in the covariance matrix or consider only a subset of parameters, in the spirit of subnetwork Laplace (Daxberger et al., 2021b). We will focus on exploiting the structure of the covariance as using a subset of parameters trivially extends from our discussion.

$$
\mathbb{C}\mathrm{ov}[\boldsymbol{W}] = \left(\begin{array}{cc|cc|cc}
W_{11}, W_{21} & W_{11}, W_{23} & W_{11}, W_{33} & W_{11}, W_{31} & W_{11}, W_{32} & W_{11}, W_{33} \\
W_{12}, W_{11} & W_{12}, W_{12} & W_{12}, W_{13} & W_{12}, W_{31} & W_{12}, W_{32} & W_{12}, W_{33} \\
\hline
W_{13}, W_{11} & W_{13}, W_{13} & W_{13}, W_{13} & W_{13}, W_{31} & W_{13}, W_{32} & W_{13}, W_{33} \\
W_{21}, W_{11} & W_{21}, W_{12} & W_{21}, W_{13} & W_{21}, W_{31} & W_{21}, W_{32} & W_{21}, W_{33} \\
\hline
W_{22}, W_{11} & W_{22}, W_{12} & W_{22}, W_{13} & W_{22}, W_{31} & W_{22}, W_{32} & W_{22}, W_{33} \\
W_{23}, W_{11} & W_{23}, W_{12} & W_{23}, W_{13} & W_{23}, W_{31} & W_{23}, W_{32} & W_{23}, W_{33}
\end{array}\right)
$$

Figure 3: To retrieve the highlighted submatrix $\mathbb{C}\mathrm{ov}[\boldsymbol{W}[1,:], \boldsymbol{W}[2,:]]$ of the covariance for $\boldsymbol{W} \in \mathbb{R}^{2 \times 3}$, we identify the Kronecker blocks that contain the covariance of interest (II, III, V, and VI), explicate those blocks in memory, and then retrieve the relevant submatrix.

**Diagonal approximation** In case the correlations between model parameters are ignored, as in mean-field variational inference, the computation of the pre-activation covariance reduces to:

$$
\mathbb{C}\mathrm{ov}\left[h_k^{(m)}, h_l^{(m)}\right] = \sum_{1 \leq i,j \leq D_{\text{in}}} \mathbb{E}\left[W_{ki}^{(m)}\right] \mathbb{E}\left[W_{lj}^{(m)}\right] \mathbb{C}\mathrm{ov}\left[a_i^{(m-1)}, a_j^{(m-1)}\right], \tag{8}
$$

and variance of the $k^{\text{th}}$ pre-activation is given as: $\mathbb{V}\mathrm{ar}[h_k^{(m)}] =$

$$
\sum_{1 \leq i \leq D_{\text{in}}} \mathbb{E}\left[a_i^{(m)}\right]^2 \mathbb{V}\mathrm{ar}\left[W_{ki}^{(m)}\right] + \mathbb{V}\mathrm{ar}\left[b_k^{(m)}\right] + \mathbb{V}\mathrm{ar}\left[a_i^{(m-1)}\right] \left(\mathbb{E}\left[W_{ki}^{(m)}\right]^2 + \mathbb{V}\mathrm{ar}\left[W_{ki}^{(m)}\right]\right). \tag{9}
$$

Hence, assuming a diagonal covariance structure can help in reducing the computational burden. If only the variance of the layer output is of interest, the computational cost can be further reduced and adds to a total of $\mathcal{O}(\sum_{m=1}^{M} D_{\text{out}}^{(m)} D_{\text{in}}^{(m)} + D_{\text{out}}^{(m)})$. Further details are given in App. A.3. An empirical run time analysis indicating little to no overhead is given in App. B.7.

**Kronecker-factorisation (KFAC)** Another common choice for approximating the posterior covariance is the use of a Kronecker-factorisation (KFAC) (Martens & Grosse, 2015), popularised in the context of Laplace approximations (Ritter et al., 2018). In this case, the posterior covariance $\boldsymbol{\Sigma}$ is given by a Kronecker product of two factors, *i.e.*, $\boldsymbol{\Sigma} = (\boldsymbol{A} \otimes \boldsymbol{B} + \lambda^2 \mathbf{I})^{-1}$ where $\otimes$ denotes the Kronecker product and $\lambda^2 \mathbf{I}$ is a prior precision. Note that in case of a non-zero prior precision, the covariance cannot be expressed in the form of a Kronecker matrix multiplication. Denote the eigenvectors and eigenvalues of $\boldsymbol{A}$ as $\boldsymbol{U}_A$ and $\boldsymbol{\Lambda}_A$, respectively, we approximate the $\boldsymbol{\Sigma}$ as follows:

$$
\boldsymbol{\Sigma} = (\boldsymbol{A} \otimes \boldsymbol{B} + \lambda^2 \mathbf{I})^{-1} = \left((\boldsymbol{U}_A \boldsymbol{\Lambda}_A \boldsymbol{U}_A^\top) \otimes (\boldsymbol{U}_B \boldsymbol{\Lambda}_B \boldsymbol{U}_B^\top) + \lambda^2 \mathbf{I}\right)^{-1} \qquad \text{(Eigen Decomposition)}
$$

$$
\approx \left((\boldsymbol{U}_A \otimes \boldsymbol{U}_B)(\boldsymbol{\Lambda}_A + \lambda \mathbf{I}_A)\right)^{-1} \otimes \left((\boldsymbol{\Lambda}_B + \lambda \mathbf{I}_B)(\boldsymbol{U}_A \otimes \boldsymbol{U}_B)^\top\right)^{-1}. \tag{10}
$$

To compute the covariance of the pre-activations, we need to retrieve the covariance between the weights of the $k^{\text{th}}$ unit and the weights of the $l^{\text{th}}$ unit, which corresponds to the $k^{\text{th}}$ and $l^{\text{th}}$ row in $\boldsymbol{W}$, *i.e.*, $\mathbb{C}\mathrm{ov}[\boldsymbol{W}[k,:], \boldsymbol{W}[l,:]]$. In case of KFAC Laplace approximations, accessing $\mathbb{C}\mathrm{ov}[\boldsymbol{W}[k,:], \boldsymbol{W}[l,:]]$ cannot be done directly. Therefore, we developed a block retrieval method to retrieve $\mathbb{C}\mathrm{ov}[\boldsymbol{W}[k,:], \boldsymbol{W}[l,:]]$ without explicating the full covariance matrix in memory.

The key idea is to first identify the Kronecker blocks that contain the covariance of interest and then retrieve the submatrix by reconstructing only the relevant blocks. Fig. 3 illustrates the idea in a toy example for a weight matrix $\boldsymbol{W} \in \mathbb{R}^{2 \times 3}$ and a submatrix of interest $\mathbb{C}\mathrm{ov}[\boldsymbol{W}[1,:], \boldsymbol{W}[2,:]]$. Our method only reconstructs blocks containing the sub-covariance of interest (II, III, V, and VI) and then retrieves the relevant submatrix. Further details are given in App. A.4.

## 4 EXPERIMENTS

We demonstrate *practical applicability* of our approach on classification/regression tasks (Sec. 4.1), large-scale classification results with ViT/GPT models (Sec. 4.2), and sensitivity estimation (Sec. 4.3). Additional experiments and additional experimental results can be found in App. B.

**Data sets** We use a selection of data sets from the UCI repository (Kelly et al., 2023) for the regression experiments. For classification, we experiment on MNIST (LeCun et al., 1998), FMNIST (Xiao et al.,

2017), as well as the 11-class data sets OrganCMNIST and OrganSMNIST from MedMNIST (Yang et al., 2023). To assess our method on higher-dimensional settings, we experiment with CIFAR-10 and CIFAR-100 (Krizhevsky & Hinton, 2009), DTD (Cimpoi et al., 2014), RESISC (Cheng et al., 2017) and a subsampled version of ImageNet-R (Hendrycks et al., 2021) with 100 classes to reduce the memory overhead for the LA. For the GPT model, we used the BOOLQ, WIC, and MRPC tasks from GLUE (Wang et al., 2019b) and SuperGLUE (Wang et al., 2019a) benchmarks.

**Posterior approximations** We adopt the Laplace approximation (LA, MacKay (1996)) and mean-field variational inference (MFVI, Blei et al., 2017) for approximating the posterior distribution of the network parameters. For the LA, we estimate the full covariance for the regression experiments, while we use diagonal or KFAC approximations for the covariance where applicable in the classification experiments. We compare our method using local Gaussian approximation and local linearisation against Monte Carlo (MC) sampling and a global linearised model (GLM, Immer et al., 2021b). For MFVI, we adopt the IVON optimiser (Shen et al., 2024) to obtain the posterior approximation with a diagonal covariance structure by default, which has been shown to be effective and scalable to large-scale classification tasks. Here, we compare our method against MC sampling from the posterior to make predictions as done in Shen et al. (2024). For the MFVI and LA sampling baselines, we used $1,000$ MC samples in the regression and classification experiments in Sec. 4.1, and 50 MC samples for the ViT and GPT-2 in Sec. 4.2. For our method, we fit an additional scaling factor on the predictive variance by minimising the NLPD on a validation set, similar to the pseudo-count used in Ritter et al. (2018).

**Network architectures** We experiment with one or two-layer multi-layer perceptron (MLP) on the UCI regression data sets with details given in App. B.1. For MNIST, FMNIST, OrganCMNIST and OrganSMNIST, we use an MLP with layers containing $784 - 128 - 64 - C$ neurons, where $C$ is the number of classes. For CIFAR-10/100, DTD, RESISC and ImageNet-R, we fine-tune a Vision Transformer (ViT) (Dosovitskiy et al., 2021) base model pre-trained on ImageNet-1k (Deng et al., 2009). For the GPT model, we use the pre-trained GPT-2 base model from Hugging Face Transformers (Wolf et al., 2019) and fine-tune it on the respective tasks.

**Evaluation metrics** For the regression experiments, we measure the negative log predictive density (NLPD) and root-mean-square error (RMSE) for each method. In the classification experiments, we use accuracy (ACC), NLPD, and expected calibration error (ECE) to compare the methods. We use a paired $t$-test with $p = 0.05$ to bold results with significant statistical differences when reporting the results. For assessing out-of-distribution (OOD) robustness, we use a Gaussian kernel density estimator with a variance of 0.25 on the histogram of the predictive entropy evaluated on the test set.

## 4.1 DOES OUR METHOD PROVIDE USEFUL UNCERTAINTY ESTIMATES?

**Regression** We experiment on a selection of data sets from the UCI repository and run a 5-fold cross validation to report results for each data set. We use either MFVI or LA to obtain the posterior approximation and separately compare our method against their corresponding prediction approaches. Table 1 shows our method achieves better NLPD in general than the predictions with sampling for both MFVI and LA. Moreover, our method performs on par with the GLM, even though our method results in a locally linearised network w.r.t. the inputs. Similar conclusions are made inspecting Table 8.

Table 1: Negative log predictive density ↓ on UCI regression data sets. Ours results in better or matching performance compared with sampling and GLM, indicating the effectiveness of our method.

| | | MFVI (Diagonal Covariance) | | Laplace Approximation (Full Covariance) | | |
| --- | --- | --- | --- | --- | --- | --- |
| | $(n, d)$ | Sampling | Ours | Sampling | GLM | Ours |
| SERVO | (167, 4) | $\mathbf{1.287}_{\pm 0.069}$ | $\mathbf{1.136}_{\pm 0.182}$ | $3.795_{\pm 0.110}$ | $\mathbf{1.047}_{\pm 0.172}$ | $1.443_{\pm 0.077}$ |
| LD | (345, 5) | $\mathbf{1.346}_{\pm 0.280}$ | $\mathbf{1.369}_{\pm 0.440}$ | $2.221_{\pm 0.110}$ | $\mathbf{1.495}_{\pm 0.580}$ | $\mathbf{1.474}_{\pm 0.648}$ |
| AM | (398, 7) | $1.004_{\pm 0.052}$ | $\mathbf{0.807}_{\pm 0.087}$ | $1.812_{\pm 0.065}$ | $\mathbf{0.492}_{\pm 0.279}$ | $\mathbf{0.478}_{\pm 0.309}$ |
| REV | (414, 6) | $1.076_{\pm 0.059}$ | $\mathbf{0.925}_{\pm 0.091}$ | $1.932_{\pm 0.045}$ | $0.859_{\pm 0.129}$ | $\mathbf{0.833}_{\pm 0.156}$ |
| FF | (517, 12) | $\mathbf{2.160}_{\pm 3.003}$ | $\mathbf{2.333}_{\pm 3.671}$ | $2.086_{\pm 0.292}$ | $\mathbf{1.584}_{\pm 0.950}$ | $\mathbf{1.596}_{\pm 1.217}$ |
| ITT | (1020, 33) | $0.937_{\pm 0.047}$ | $\mathbf{0.841}_{\pm 0.065}$ | $1.681_{\pm 0.069}$ | $0.825_{\pm 0.095}$ | $\mathbf{0.756}_{\pm 0.164}$ |
| CCS | (1030, 8) | $0.939_{\pm 0.068}$ | $\mathbf{0.828}_{\pm 0.108}$ | $1.612_{\pm 0.048}$ | $0.319_{\pm 0.109}$ | $\mathbf{0.234}_{\pm 0.161}$ |
| ASN | (1503, 5) | $0.962_{\pm 0.054}$ | $\mathbf{0.899}_{\pm 0.065}$ | $1.788_{\pm 0.045}$ | $0.422_{\pm 0.109}$ | $\mathbf{0.396}_{\pm 0.133}$ |
| CAC | (1994, 127) | $0.973_{\pm 0.092}$ | $\mathbf{0.920}_{\pm 0.118}$ | $1.848_{\pm 0.055}$ | $\mathbf{1.281}_{\pm 0.069}$ | $2.662_{\pm 1.096}$ |
| PT | (5875, 19) | $0.976_{\pm 0.069}$ | $\mathbf{0.940}_{\pm 0.074}$ | $0.984_{\pm 0.101}$ | $\mathbf{0.576}_{\pm 0.181}$ | $0.651_{\pm 0.306}$ |
| CCPP | (9568, 4) | $0.365_{\pm 0.040}$ | $\mathbf{0.352}_{\pm 0.042}$ | $1.345_{\pm 0.085}$ | $\mathbf{-0.062}_{\pm 0.182}$ | $\mathbf{-0.062}_{\pm 0.200}$ |
| Bold Count | | 3/11 | 11/11 | 0/11 | 7/11 | 8/11 |

Table 2: Performance metrics on the MNIST-like data sets for each method with the standard error for ACC and NLPD. Our method achieves better NLPD and ECE than the baselines.

| Metrics | Methods | MNIST | FMNIST | ORGANCMNIST | ORGANSMNIST |
|---|---|---|---|---|---|
| ACC ↑ | LA Sampling | $0.972_{\pm0.002}$ | $0.868_{\pm0.004}$ | $0.734_{\pm0.008}$ | $0.591_{\pm0.010}$ |
| | LA GLM | $\mathbf{0.975}_{\pm0.002}$ | $\mathbf{0.882}_{\pm0.003}$ | $\mathbf{0.824}_{\pm0.007}$ | $\mathbf{0.634}_{\pm0.009}$ |
| | LA Ours | $\mathbf{0.975}_{\pm0.002}$ | $0.881_{\pm0.003}$ | $\mathbf{0.826}_{\pm0.008}$ | $0.630_{\pm0.009}$ |
| | MFVI Sampling | $0.974_{\pm0.002}$ | $0.843_{\pm0.004}$ | $0.620_{\pm0.010}$ | $0.467_{\pm0.010}$ |
| | MFVI Ours | $0.974_{\pm0.002}$ | $0.842_{\pm0.004}$ | $\mathbf{0.630}_{\pm0.010}$ | $0.454_{\pm0.010}$ |
| NLPD ↓ | LA Sampling | $0.210_{\pm0.003}$ | $0.556_{\pm0.008}$ | $1.135_{\pm0.017}$ | $1.614_{\pm0.021}$ |
| | LA GLM | $\mathbf{0.089}_{\pm0.004}$ | $0.548_{\pm0.018}$ | $0.875_{\pm0.44}$ | $1.967_{\pm0.070}$ |
| | LA Ours | $\mathbf{0.089}_{\pm0.005}$ | $\mathbf{0.397}_{\pm0.010}$ | $\mathbf{0.710}_{\pm0.021}$ | $\mathbf{1.365}_{\pm0.025}$ |
| | MFVI Sampling | $0.179_{\pm0.014}$ | $2.010_{\pm0.051}$ | $4.775_{\pm0.140}$ | $6.095_{\pm0.141}$ |
| | MFVI Ours | $\mathbf{0.086}_{\pm0.005}$ | $\mathbf{0.529}_{\pm0.011}$ | $\mathbf{1.492}_{\pm0.026}$ | $\mathbf{1.895}_{\pm0.022}$ |
| ECE ↓ | LA Sampling | 0.122 | 0.151 | 0.292 | 0.261 |
| | LA GLM | 0.009 | 0.078 | $\mathbf{0.038}$ | 0.170 |
| | LA Ours | $\mathbf{0.004}$ | $\mathbf{0.012}$ | 0.122 | $\mathbf{0.151}$ |
| | MFVI Sampling | 0.018 | 0.144 | 0.251 | 0.385 |
| | MFVI Ours | $\mathbf{0.003}$ | $\mathbf{0.013}$ | $\mathbf{0.145}$ | $\mathbf{0.067}$ |

Table 3: Performance comparison between our method and Moment-Matching (MM) on the MNIST-like classification tasks. We report the ACC and NLPD with standard errors and the ECE. Our method outperforms MM despite being simpler and more applicable to various distributions.

| Metrics | Methods | MNIST | FMNIST | ORGANCMNIST | ORGANSMNIST |
|---|---|---|---|---|---|
| ACC ↑ | MM | $0.971_{\pm0.002}$ | $\mathbf{0.882}_{\pm0.003}$ | $0.771_{\pm0.004}$ | $0.600_{\pm0.010}$ |
| | Ours | $\mathbf{0.975}_{\pm0.002}$ | $0.881_{\pm0.003}$ | $\mathbf{0.816}_{\pm0.004}$ | $\mathbf{0.614}_{\pm0.010}$ |
| NLPD ↓ | MM | $0.103_{\pm0.005}$ | $0.463_{\pm0.014}$ | $0.838_{\pm0.015}$ | $1.401_{\pm0.036}$ |
| | Ours | $\mathbf{0.090}_{\pm0.005}$ | $\mathbf{0.435}_{\pm0.010}$ | $\mathbf{0.733}_{\pm0.010}$ | $\mathbf{1.282}_{\pm0.024}$ |
| ECE ↓ | MM | 0.014 | 0.051 | $\mathbf{0.023}$ | 0.110 |
| | Ours | $\mathbf{0.006}$ | $\mathbf{0.022}$ | 0.127 | $\mathbf{0.081}$ |

**Classification** Here, we assess our method on MNIST-like classification tasks. For the LA, we use KFAC approximation of the covariance to reduce the memory overhead. In Table 2, we report the ACC and NLPD with their standard errors and the ECE for each method. Our method achieves similar ACC with the baselines, while outperforming them on the NLPD and ECE metrics. In App. B.2, we assess our method on robustness to OOD data. We evaluate an MLP trained on MNIST on rotated versions of the test set. Our method consistently reduces overconfidence on OOD data, *cf.*, Fig. 8.

**Ours vs. moment-matching** To verify the viability of local linearisation, we compare our method against moment-matching (MM) used in Wu et al. (2019). We apply MM instead of local linearisation to our setting, assuming a diagonal posterior approximation from LA. In Table 3, we show the results for our method against MM on MNIST-like classification tasks. Our approach outperforms MM across the data sets and the metrics, except for the ECE on OrganCMNIST. Compared to MM, our method is applicable to any differentiable activation function and any type of stable distribution (Petersen et al., 2024), while MM requires tailored derivations for each case and, hence, is less *plug-and-play*.

### 4.2 IS OUR METHOD SCALABLE?

We demonstrate that our method is applicable to large-scale networks by experimenting with pre-trained ViT and GPT-2 models. In particular, we experiment with applying our method on either the attention layer or the MLP after the attention layer in the last two (ViT) / four (GPT) transformer blocks. For each target data set, we fine-tune the layers we obtain the posterior approximation for. We assume a diagonal posterior to reduce the memory overhead. Table 4 shows the results when fine-tuning ViT models and obtaining the posterior approximation from the attention layers. We observe that our method achieves better or on par NLPD and ECE compared to the baselines for both LA and MFVI across all data sets while maintaining similar ACC as the baselines.

In Table 5 we show the results for a GPT-2 model with LA on the MLP layers. We observe that our method systematically outperforms sampling, while achieving similar performance to GLM in some cases. Thus indicating that our method is applicable to different application domains. We present additional results for ViT models in App. B.

We also assess the robustness to out-of-distribution (OOD) data for our method and the baselines. In particular, we take the ViT network fine-tuned on CIFAR-10 and evaluate its predictive entropy on

Table 4: Performance metrics using ViT with posterior approximation on the attention layers with the standard error for ACC and NLPD. Our method achieves better NLPD and ECE in general and achieves similar ACC compared to the baselines.

| Metrics | Methods | CIFAR-10 | CIFAR-100 | DTD | RESISC | IMAGENET-R |
|---------|---------|----------|-----------|-----|--------|------------|
| ACC ↑ | LA Sampling | $0.971_{\pm 0.002}$ | $\mathbf{0.882}_{\pm 0.003}$ | $0.715_{\pm 0.010}$ | $\mathbf{0.892}_{\pm 0.004}$ | $0.731_{\pm 0.012}$ |
| | LA GLM | $\mathbf{0.976}_{\pm 0.002}$ | $0.879_{\pm 0.003}$ | $0.718_{\pm 0.010}$ | $0.891_{\pm 0.004}$ | $\mathbf{0.739}_{\pm 0.012}$ |
| | LA Ours | $\mathbf{0.976}_{\pm 0.002}$ | $0.880_{\pm 0.003}$ | $0.719_{\pm 0.010}$ | $\mathbf{0.892}_{\pm 0.004}$ | $\mathbf{0.739}_{\pm 0.012}$ |
| | MFVI Sampling | $0.975_{\pm 0.002}$ | $0.880_{\pm 0.003}$ | $0.732_{\pm 0.010}$ | $0.867_{\pm 0.004}$ | $\mathbf{0.730}_{\pm 0.012}$ |
| | MFVI Ours | $0.975_{\pm 0.002}$ | $0.880_{\pm 0.003}$ | $\mathbf{0.734}_{\pm 0.010}$ | $0.867_{\pm 0.004}$ | $0.727_{\pm 0.012}$ |
| NLPD ↓ | LA Sampling | $0.170_{\pm 0.004}$ | $\mathbf{0.444}_{\pm 0.012}$ | $1.238_{\pm 0.028}$ | $0.461_{\pm 0.009}$ | $1.208_{\pm 0.048}$ |
| | LA GLM | $0.092_{\pm 0.007}$ | $0.459_{\pm 0.012}$ | $1.197_{\pm 0.029}$ | $0.385_{\pm 0.010}$ | $\mathbf{1.180}_{\pm 0.047}$ |
| | LA Ours | $\mathbf{0.086}_{\pm 0.006}$ | $0.456_{\pm 0.012}$ | $\mathbf{1.068}_{\pm 0.035}$ | $\mathbf{0.353}_{\pm 0.012}$ | $1.264_{\pm 0.043}$ |
| | MFVI Sampling | $0.133_{\pm 0.011}$ | $0.641_{\pm 0.022}$ | $1.091_{\pm 0.048}$ | $1.010_{\pm 0.041}$ | $1.577_{\pm 0.083}$ |
| | MFVI Ours | $\mathbf{0.087}_{\pm 0.006}$ | $0.468_{\pm 0.012}$ | $\mathbf{1.006}_{\pm 0.035}$ | $0.619_{\pm 0.019}$ | $\mathbf{1.234}_{\pm 0.052}$ |
| ECE ↓ | LA Sampling | $\mathbf{0.006}$ | $\mathbf{0.022}$ | $0.197$ | $0.129$ | $0.070$ |
| | LA GLM | $0.011$ | $0.024$ | $0.155$ | $0.053$ | $\mathbf{0.057}$ |
| | LA Ours | $0.009$ | $0.027$ | $\mathbf{0.042}$ | $\mathbf{0.017}$ | $0.130$ |
| | MFVI Sampling | $0.015$ | $0.070$ | $0.075$ | $0.079$ | $0.118$ |
| | MFVI Ours | $\mathbf{0.007}$ | $\mathbf{0.024}$ | $\mathbf{0.040}$ | $\mathbf{0.018}$ | $\mathbf{0.043}$ |

Table 5: Performance on language understanding tasks.

| Metrics | Methods | BOOLQ | WIC | MRPC |
|---------|---------|-------|-----|------|
| ACC ↑ | LA Sampling | $0.383_{\pm 0.009}$ | $0.500_{\pm 0.020}$ | $0.335_{\pm 0.011}$ |
| | LA GLM | $\mathbf{0.698}_{\pm 0.008}$ | $\mathbf{0.611}_{\pm 0.019}$ | $\mathbf{0.710}_{\pm 0.011}$ |
| | LA Ours | $0.641_{\pm 0.008}$ | $0.500_{\pm 0.020}$ | $0.665_{\pm 0.011}$ |
| NLPD ↓ | LA Sampling | $0.814_{\pm 0.006}$ | $0.858_{\pm 0.024}$ | $1.219_{\pm 0.018}$ |
| | LA GLM | $\mathbf{0.650}_{\pm 0.014}$ | $0.730_{\pm 0.027}$ | $0.660_{\pm 0.023}$ |
| | LA Ours | $0.684_{\pm 0.001}$ | $\mathbf{0.719}_{\pm 0.010}$ | $\mathbf{0.629}_{\pm 0.009}$ |
| ECE ↓ | LA Sampling | $0.259$ | $0.261$ | $0.484$ |
| | LA GLM | $\mathbf{0.086}$ | $0.126$ | $0.145$ |
| | LA Ours | $0.129$ | $\mathbf{0.118}$ | $\mathbf{0.054}$ |

the SVHN data set (Netzer et al., 2011). Fig. 4 shows the kernel density of the predictive entropy computed on the test sets of CIFAR-10 and SVHN, where the model should have high entropy for data different from the training data. Although our method is slightly underconfident on the in-distribution data, the entropy for in-distribution and OOD data is clearly separated, especially for MFVI.

### 4.3 CAN OUR METHOD ESTIMATE INPUT SENSITIVIES?

We demonstrate that our method can estimate sensitivities w.r.t. the inputs to the network. For this, we use a 3-class MLP trained on the digits $0/6/8$. Our goal is to estimate sensitivity maps by assuming that the input images $\boldsymbol{x} \sim \mathcal{N}(\boldsymbol{x}, \boldsymbol{\Sigma})$ are distributed according to a Gaussian centred at the pixel values with diagonal covariance $\boldsymbol{\Sigma}$. We optimise the input covariance of each image by minimising the loss

$$\ell = \sum_{n=1}^{N} \text{cross-entropy}(f(\boldsymbol{x}_n), y_n) - \mathcal{H}(\mathcal{N}(\boldsymbol{x}_n, \Sigma_n)). \tag{11}$$

In words, we jointly minimise the cross-entropy loss, after analytically propagating the input distribution through the network, while maximising the entropy $\mathcal{H}(\mathcal{N}(\boldsymbol{x}_n, \Sigma_n))$ of the input distribution. The optimisation is stopped once the difference in NLPD between the current iteration and initial condition is more than $0.1$. Fig. 5 shows examples of the resulting sensitivity maps for a deterministic MLP (MAP) and the same MLP with last-layer LA (Bayes). We observe that the largest sensitivity for the digits $0$ and $8$ are generally in the middle, while for $6$ in the upper right corner. The Bayes model shows less spurious sensitivities across the pixels compared to the MAP model. Thus, indicating that incorporating all sources of uncertainties can lead to a more interpretable sensitivity analysis.

## 5 DISCUSSION & CONCLUSION

In this work, we proposed to streamline prediction in Bayesian deep learning through local linearisation and local Gaussian approximations of the network. For this, we discussed the propagation in different neural network architectures and covariance structures. In particular, we discussed how to handle Kronecker-factorised posterior covariances and transformer architectures. We showed through a series of experiments that our method obtains high predictive performance, provides useful predictive uncertainties, and can be used for sensitivity analysis. Our method helps to make BDL more useful in practice and expands the use cases and sources of uncertainties that can be considered.

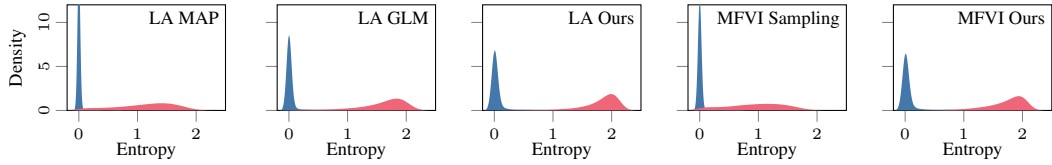

Figure 4: Kernel density plots over the predictive entropy from a ViT network finetuned on CIFAR-10 (blue, in-distribution) and data from SVHN (red, out-of-distribution). Our method results in a clear separation between the in- and out-of-distribution data.

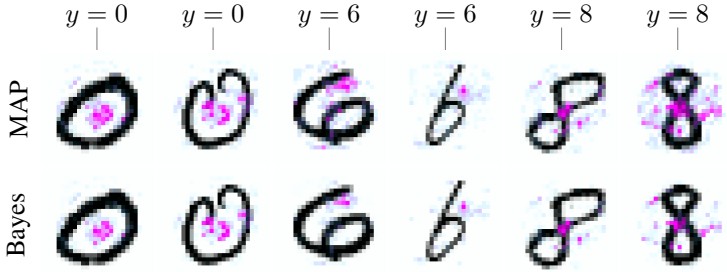

Figure 5: Pixel sensitivity maps of an MLP trained on a subset of MNIST digits (classes $0/6/8$). The rows show sensitivities to pixel perturbations for the MLP (MAP) and MLP with last-layer Laplace approximation (Bayes) respectively. The sensitivities are visualised in the range $(0.5 \quad 1.0)$. The Bayes MLP shows less spurious sensitivities across the pixels compared to MAP.

In future work, we aim to apply our approach to tasks with a larger number of output classes, explore additional use-case scenarios in which our streamlined approach can be beneficial, and scale to even larger networks. Moreover, we aim to investigate further the computational benefits obtained by exploiting the posterior covariance structure and sparsity in the network.

**Limitations** The local linearisation of activation functions induces an error that depends on both the activation function and the location and scale of the distribution over the input to the activation function. Moreover, we assume independence between the activations and model parameters for the local Gaussian approximation in linear layers and residual connections, which may incur a loss of information in the propagation. Especially, the independence assumption in the residual block is potentially harmful, and relaxing it would be a valuable future direction. Further, it would be interesting to estimate the induced approximation error to identify potential failure modes. Finally, we assume access to a validation set for fitting the scaling factor of the predictive posterior distribution, which is currently done using a grid search. An interesting future step is the use of the marginal likelihood to optimise the scaling factor.

## ACKNOWLEDGMENTS

AS and RL acknowledge funding from the Research Council of Finland (grant number 339730, 362408). MT acknowledges funding from the Research Council of Finland (grant number 347279). MK acknowledges funding from the Finnish Center for Artificial Intelligence (FCAI). We acknowledge CSC – IT Center for Science, Finland, for awarding this project access to the LUMI supercomputer, owned by the EuroHPC Joint Undertaking, hosted by CSC (Finland) and the LUMI consortium through CSC. We acknowledge the computational resources provided by the Aalto Science-IT project. Lastly, we thank Jonas Vestergaard for finding a bug in our code and his feedback on the manuscript.

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

## APPENDICES

The appendices are structured as follows: App. A presents the derivations of our method in detail. App. B describes the experimental setup and additional experimental results.

## A    DERIVATIONS

In this section, we derive how to propagate the distribution deterministically. See Table 6 for the list of notations that will be used throughout this section.

We first derive the general result in App. A.1 where the posterior covariance has a full structure in the linear layer and evaluate the quality of local Gaussian approximation in App. A.2. Next, in App. A.3 and App. A.4 we give the derivation for diagonal and KFAC covariance, respectively. Then, App. A.5 shows the derivation for activation functions. Finally, App. A.6 describes how we apply our method in a transformer network (Vaswani et al., 2017).

Table 6: Notation.

| | |
|---|---|
| $\boldsymbol{x}$ | lowercase bolder letter, vector |
| $\boldsymbol{W}$ | uppercase bold letter, matrix |
| $\mathcal{D}$ | set |
| $x_i$ | $i^{\text{th}}$ element of $\boldsymbol{x}$ |
| $W_{ki}$ | $k^{\text{th}}$ row, $i^{\text{th}}$ column of $\boldsymbol{W}$ |
| $\boldsymbol{W}[k,:]$ | $k^{\text{th}}$ row of a matrix |
| $k, l$ | dimension of the output |
| $i, j$ | dimension of the input |
| $d$ | data feature dimension |
| $n, N$ | number of data points |
| $C$ | total number of classes |
| $m$ | layer index |

### A.1    DERIVATION FOR FULL COVARIANCE STRUCTURE

Denote the weight and bias of the $m^{\text{th}}$ linear layer as $\boldsymbol{W}^{(m)} \in \mathbb{R}^{D_{\text{out}} \times D_{\text{in}}}$ and $\boldsymbol{b}^{(m)} \in \mathbb{R}^{D_{\text{out}}}$ respectively, and its input as $\boldsymbol{a}^{(m-1)} \in \mathbb{R}^{D_{\text{in}}}$. The pre-activation is then given as $\boldsymbol{h}^{(m)} = \boldsymbol{W}^{(m)} \boldsymbol{a}^{(m-1)} + \boldsymbol{b}^{(m)}$ with its $k^{\text{th}}$ element being $h_k^{(m)} = \sum_{i=1}^{D_{\text{in}}} W_{ki}^{(m)} a_i^{(m-1)} + b_k^{(m)}$.

We make the following assumptions to obtain a tractable distribution on the pre-activation:

- Assumption 1: We assume each $a_i^{(m-1)} W_{ki}^{(m)}$ is a Gaussian distribution.
- Assumption 2: We assume that the activations of the previous layer $a_i^{(m-1)}$ and parameters of the $m^{\text{th}}$ layer are independent.

From assumption 1, because now $a_i^{(m-1)} W_{ki}^{(m)}$ and $b_k^{(m)}$ are all Gaussian distributions, $h_k^{(m)}$ will follow Gaussian distribution as well. We call this local Gaussian approximation as we approximate each local component $a_i^{(m-1)} W_{ki}^{(m)}$ with a Gaussian. As now each $h_k^{(m)}$ is a Gaussian, $\boldsymbol{h}^{(m)}$ will be jointly Gaussian. We derive its mean and covariance and drop the layer index if it is clear from the context.

**Derivation of mean** As $a_i$ is assumed to be uncorrected with $W_{ki}$, we have

$$\mathbb{E}[h_k] = \mathbb{E}\left[\sum_{i=1}^{D_{\text{in}}} W_{ki}a_i + b_k\right] \tag{12}$$

$$= \sum_{i=1}^{D_{\text{in}}} \mathbb{E}[W_{ki}a_i + b_k] \tag{13}$$

$$= \sum_{i=1}^{D_{\text{in}}} \mathbb{E}[W_{ki}a_i] + \mathbb{E}[b_k] \tag{14}$$

$$\approx \sum_{i=1}^{D_{\text{in}}} \mathbb{E}[W_{ki}]\mathbb{E}[a_i] + \mathbb{E}[b_k]. \tag{Assumption 2}$$

**Derivation of covariance** The covariance between the $k^{\text{th}}$ and $l^{\text{th}}$ pre-activation can be written as

$$\mathbb{C}\text{ov}[h_k, h_l] = \mathbb{C}\text{ov}\left[\sum_{i=1}^{D_{\text{in}}} a_i W_{ki} + b_k, \sum_{i=1}^{D_{\text{in}}} a_i W_{li} + b_l\right] \tag{15}$$

$$= \mathbb{C}\text{ov}\left[\sum_{i=1}^{D_{\text{in}}} a_i W_{ki}, \sum_{i=1}^{D_{\text{in}}} a_i W_{li}\right] + \mathbb{C}\text{ov}\left[\sum_{i=1}^{D_{\text{in}}} a_i W_{ki}, b_l\right] + \mathbb{C}\text{ov}\left[\sum_{i=1}^{D_{\text{in}}} a_i W_{li}, b_k\right]$$
$$+ \mathbb{C}\text{ov}[b_k, b_l] \tag{16}$$

$$= \sum_{1 \leq i,j \leq D_{\text{in}}} \mathbb{C}\text{ov}[a_i W_{ki}, a_j W_{lj}] + \sum_{1 \leq i \leq D_{\text{in}}} (\mathbb{C}\text{ov}[a_i W_{ki}, b_l] + \mathbb{C}\text{ov}[a_i W_{li}, b_k])$$
$$+ \mathbb{C}\text{ov}[b_k, b_l] \tag{17}$$

We first derive the form of $\mathbb{C}\text{ov}[a_i W_{ki}, a_i W_{li}]$:

$\mathbb{C}\text{ov}[a_i W_{ki}, a_j W_{lj}]$

$$= \mathbb{E}[(a_i W_{ki} - \mathbb{E}[a_i W_{ki}])(a_j W_{lj} - \mathbb{E}[a_j W_{lj}])] \tag{18}$$

$$= \mathbb{E}[a_i W_{ki} a_j W_{lj} - a_i W_{ki}\mathbb{E}[a_j W_{lj}] - \mathbb{E}[a_i W_{ki}] a_j W_{lj} + \mathbb{E}[a_i W_{ki}]\mathbb{E}[a_j W_{lj}]] \tag{19}$$

$$= \mathbb{E}[a_i a_j W_{ki} W_{lj}] - \mathbb{E}[a_i W_{ki}]\mathbb{E}[a_j W_{lj}] - \mathbb{E}[a_i W_{ki}]\mathbb{E}[a_j W_{lj}] + \mathbb{E}[a_i W_{ki}]\mathbb{E}[a_j W_{lj}] \tag{20}$$

$$\approx \mathbb{E}[a_i a_j]\mathbb{E}[W_{ki} W_{lj}] - \mathbb{E}[a_i]\mathbb{E}[W_{ki}]\mathbb{E}[a_j]\mathbb{E}[W_{lj}] \tag{Assumption 2}$$

$$= (\mathbb{E}[a_i]\mathbb{E}[a_j] + \mathbb{C}\text{ov}[a_i, a_j])(\mathbb{E}[W_{ki}]\mathbb{E}[W_{lj}] + \mathbb{C}\text{ov}[W_{ki}, W_{lj}])$$
$$- \mathbb{E}[a_i]\mathbb{E}[W_{ki}]\mathbb{E}[a_j]\mathbb{E}[W_{lj}] \tag{21}$$

$$= \mathbb{E}[a_i]\mathbb{E}[a_j]\mathbb{C}\text{ov}[W_{ki}, W_{lj}] + \mathbb{E}[W_{ki}]\mathbb{E}[W_{lj}]\mathbb{C}\text{ov}[a_i, a_j] + \mathbb{C}\text{ov}[a_i, a_j]\mathbb{C}\text{ov}[W_{ki}, W_{lj}]. \tag{22}$$

Then we derive the form of $\mathbb{C}\text{ov}[a_i W_{ki}, b_l]$:

$$\mathbb{C}\text{ov}[a_i W_{ki}, b_l] = \mathbb{E}[(a_i W_{ki} - \mathbb{E}[a_i W_{ki}])(b_l - \mathbb{E}[b_l])] \tag{23}$$

$$\approx \mathbb{E}[(a_i W_{ki} - \mathbb{E}[a_i]\mathbb{E}[W_{ki}])(b_l - \mathbb{E}[b_l])] \tag{Assumption 2}$$

$$= \mathbb{E}[a_i W_{ki} b_l - a_i W_{ki}\mathbb{E}[b_l] - \mathbb{E}[a_i]\mathbb{E}[W_{ki}] b_l + \mathbb{E}[a_i]\mathbb{E}[W_{ki}]\mathbb{E}[b_l]] \tag{24}$$

$$= \mathbb{E}[a_i W_{ki} b_l] - \mathbb{E}[a_i]\mathbb{E}[W_{ki}]\mathbb{E}[b_l] \tag{25}$$

$$\approx \mathbb{E}[a_i]\mathbb{E}[W_{ki} b_l] - \mathbb{E}[a_i]\mathbb{E}[W_{ki}]\mathbb{E}[b_l] \tag{Assumption 2}$$

$$= \mathbb{E}[a_i](\mathbb{E}[W_{ki}]\mathbb{E}[b_l] + \mathbb{C}\text{ov}[W_{ki}, b_l]) - \mathbb{E}[a_i]\mathbb{E}[W_{ki}]\mathbb{E}[b_l] \tag{26}$$

$$= \mathbb{E}[a_i]\mathbb{C}\text{ov}[W_{ki}, b_l]. \tag{27}$$

Putting it together, we have $\mathbb{C}\text{ov}[h_k, h_l] =$

$$\sum_{1 \le i,j \le D_{\text{in}}} \mathbb{C}\text{ov}\left[a_i W_{ki}, a_j W_{lj}\right] + \sum_{i=1}^{D_{\text{in}}} \left(\mathbb{E}\left[a_i\right] \mathbb{C}\text{ov}\left[W_{ki}, b_l\right] + \mathbb{E}\left[a_i\right] \mathbb{C}\text{ov}\left[W_{li}, b_k\right]\right) + \mathbb{C}\text{ov}\left[b_k, b_l\right], \quad (28)$$

where $\mathbb{C}\text{ov}[a_i W_{ki}, a_j W_{lj}] =$

$$\mathbb{E}\left[a_i\right] \mathbb{E}\left[a_j\right] \mathbb{C}\text{ov}\left[W_{ki}, W_{lj}\right] + \mathbb{E}\left[W_{ki}\right] \mathbb{E}\left[W_{lj}\right] \mathbb{C}\text{ov}\left[a_i, a_j\right] + \mathbb{C}\text{ov}\left[a_i, a_j\right] \mathbb{C}\text{ov}\left[W_{ki}, W_{lj}\right]. \quad (29)$$

Note that $\sum_{1 \le i,j \le D_{\text{in}}} \mathbb{C}\text{ov}[a_i W_{ki}, a_j W_{lj}]$ in Eq. (28) could be rewritten into the form of matrix multiplication for efficient implementation:

$$\sum_{1 \le i,j \le D_{\text{in}}} \mathbb{C}\text{ov}\left[a_i W_{ki}, a_j W_{lj}\right] \tag{30}$$

$$= \sum_{1 \le i,j \le D_{\text{in}}} \mathbb{E}\left[a_i\right] \mathbb{E}\left[a_j\right] \mathbb{C}\text{ov}\left[W_{ki}, W_{lj}\right] + \mathbb{E}\left[W_{ki}\right] \mathbb{E}\left[W_{lj}\right] \mathbb{C}\text{ov}\left[a_i, a_j\right] + \mathbb{C}\text{ov}\left[a_i, a_j\right] \mathbb{C}\text{ov}\left[W_{ki}, W_{lj}\right] \tag{31}$$

$$= \sum \begin{bmatrix} \mathbb{E}[a_1]\mathbb{E}[a_1]\mathbb{C}\text{ov}[W_{k1}, W_{l1}] & \dots & \mathbb{E}[a_1]\mathbb{E}[a_{D_{\text{in}}}]\mathbb{C}\text{ov}[W_{k1}, W_{lD_{\text{in}}}] \\ \vdots & \vdots & \vdots \\ \mathbb{E}[a_{D_{\text{in}}}]\mathbb{E}[a_1]\mathbb{C}\text{ov}[W_{kD_{\text{in}}}, W_{l1}] & \dots & \mathbb{E}[a_1]\mathbb{E}[a_{D_{\text{in}}}]\mathbb{C}\text{ov}[W_{kD_{\text{in}}}, W_{lD_{\text{in}}}] \end{bmatrix} \tag{32}$$

$$\odot \begin{bmatrix} \mathbb{C}\text{ov}[W_{k1}, W_{l1}] & \dots & \mathbb{C}\text{ov}[W_{k1}, W_{lD_{\text{in}}}] \\ \vdots & \vdots & \vdots \\ \mathbb{C}\text{ov}[W_{kD_{\text{in}}}, W_{l1}] & \dots & \mathbb{C}\text{ov}[W_{kD_{\text{in}}}, W_{lD_{\text{in}}}] \end{bmatrix} \tag{33}$$

$$+ \sum \begin{bmatrix} \mathbb{E}[W_{k1}]\mathbb{E}[W_{l1}] & \dots & \mathbb{E}[W_{k1}]\mathbb{E}[W_{lD_{\text{in}}}] \\ \vdots & \vdots & \vdots \\ \mathbb{E}[W_{kD_{\text{in}}}]\mathbb{E}[W_{l1}] & \dots & \mathbb{E}[W_{kD_{\text{in}}}]\mathbb{E}[W_{lD_{\text{in}}}] \end{bmatrix} \odot \begin{bmatrix} \mathbb{C}\text{ov}[a_1, a_1] & \dots & \mathbb{C}\text{ov}[a_1, a_{D_{\text{in}}}] \\ \vdots & \vdots & \vdots \\ \mathbb{C}\text{ov}[a_{D_{\text{in}}}, a_1] & \dots & \mathbb{C}\text{ov}[a_{D_{\text{in}}}, a_{D_{\text{in}}}] \end{bmatrix} \tag{34}$$

$$+ \sum \begin{bmatrix} \mathbb{C}\text{ov}[a_1, a_1] & \dots & \mathbb{C}\text{ov}[a_1, a_{D_{\text{in}}}] \\ \vdots & \vdots & \vdots \\ \mathbb{C}\text{ov}[a_{D_{\text{in}}}, a_1] & \dots & \mathbb{C}\text{ov}[a_{D_{\text{in}}}, a_{D_{\text{in}}}] \end{bmatrix} \odot \begin{bmatrix} \mathbb{C}\text{ov}[W_{k1}, W_{l1}] & \dots & \mathbb{C}\text{ov}[W_{k1}, W_{lD_{\text{in}}}] \\ \vdots & \vdots & \vdots \\ \mathbb{C}\text{ov}[W_{kD_{\text{in}}}, W_{l1}] & \dots & \mathbb{C}\text{ov}[W_{kD_{\text{in}}}, W_{lD_{\text{in}}}] \end{bmatrix} \tag{35}$$

## A.2 Error Induced Through Local Gaussian Approximation

In this section, we analyse the error induced by the local Gaussian approximation. Recall that we made these two assumptions for the derivation:

- Assumption 1: We assume $a_i^{(m-1)} W_{ki}^{(m)}$ is a Gaussian distribution.

- Assumption 2: We assume that the activations of the previous layer $a_i^{(m-1)}$ and parameters of the $m^{\text{th}}$ layer are independent.

We first examine the error induced by A2 on the moments for $a_i W_{ki}$. Given two correlated univariate Gaussian $x_1$ and $x_2$, with the joint being

$$\begin{bmatrix} x_1 \\ x_2 \end{bmatrix} \sim \mathcal{N}\left( \begin{bmatrix} \mathbb{E}[x_1] \\ \mathbb{E}[x_2] \end{bmatrix}, \begin{bmatrix} \sigma_{x_1}^2 & \mathbb{C}\text{ov}[x_1, x_2] \\ \mathbb{C}\text{ov}[x_1, x_2] & \sigma_{x_2}^2 \end{bmatrix} \right), \tag{36}$$

from Nadarajah & Pogány (2016); Kan (2008), although the distribution form of $x_1 x_2$ is no longer Gaussian and intractable, its mean and variance can be computed analytically as

$$\mathbb{E}[x_1 x_2] = \mathbb{E}[x_1]\mathbb{E}[x_2] + \mathbb{C}\text{ov}[x_1, x_2], \tag{37}$$

$$\mathbb{V}\text{ar}[x_1 x_2] = \sigma_1^2 \sigma_2^2 + \sigma_1^2 \mathbb{E}[x_2]^2 + \sigma_2^2 \mathbb{E}[x_1]^2 + (\sigma_1^2 \sigma_2^2 + 2\mathbb{E}[x_1]\mathbb{E}[x_2])\mathbb{C}\text{ov}[x_1, x_2]. \tag{38}$$

Applying the above result in our case, we have

$$\mathbb{E}[a_i W_{ki}] = \mathbb{E}[a_i]\mathbb{E}[W_{ki}] + \mathbb{C}\text{ov}[a_i, W_{ki}], \tag{39}$$

$$\mathbb{V}\text{ar}[a_i W_{ki}] = \sigma_{a_i}^2 \sigma_{W_{ki}}^2 + \sigma_{a_i}^2 \mathbb{E}[W_{ki}]^2 + \sigma_{W_{ki}}^2 \mathbb{E}[a_i]^2$$
$$+ (2\mathbb{E}[a_i]\mathbb{E}[W_{ki}] + \sigma_{a_i}^2 \sigma_{W_{ki}}^2)\mathbb{C}\text{ov}[a_i, W_{ki}]. \tag{40}$$

As $\mathbb{Cov}[a_i, W_{ki}]$ is intractable, in A2 we ignore the correlation between $a_i$ and $W_{ki}$, which results in

$$\mathbb{E}[a_i W_{ki}] \approx \mathbb{E}[a_i]\mathbb{E}[W_{ki}] + \mathbb{Cov}[a_i, W_{ki}], \tag{41}$$

$$\mathbb{Var}[a_i W_{ki}] \approx \sigma_{a_i}^2 \sigma_{W_{ki}}^2 + \sigma_{a_i}^2 \mathbb{E}[W_{ki}]^2 + \sigma_{W_{ki}}^2 \mathbb{E}[a_i]^2$$

$$+ (2\mathbb{E}[a_i]\mathbb{E}[W_{ki}] + \sigma_{a_i}^2 \sigma_{W_{ki}}^2)\mathbb{Cov}[a_i, W_{ki}]. \tag{42}$$

Note that in the case of diagonal posterior covariance, as each parameter is independent of each other, A2 holds automatically. In this case, we recover the correct mean and variance for $a_i W_{ki}$.

Now, we examine the error induced by A1 and A2 through Monte Carlo estimation. Fig. 6 provides a simulation result illustrating the error induced by the local Gaussian approximation on $a_i W_{ki}$. We plot the results for weights with the largest absolute magnitude of an MLP trained on MNIST. This approximation works well in practice but fails to capture the potential skewness of the distributions.

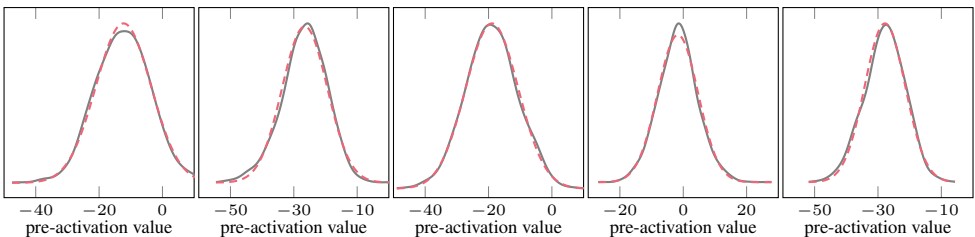

Figure 6: Comparison between Monte-Carlo estimates —— of the distribution over $a_i W_{ki}$ and our analytic Gaussian approximation - - -.

## A.3 DERIVATION FOR DIAGONAL COVARIANCE STRUCTURE

When the posterior has diagonal covariance, the mean $\mathbb{E}[h_k]$ will still be the same.

For covariance, when $k \neq l$ we have $\mathbb{Cov}[h_k, h_l] =$

$$\sum_{1 \leq i,j \leq D_{\text{in}}} \mathbb{Cov}[a_i W_{ki}, a_j W_{lj}] + \sum_{i=1}^{D_{\text{in}}} \left(\mathbb{E}[a_i]\mathbb{Cov}[W_{ki}, b_l] + \mathbb{E}[a_i]\mathbb{Cov}[W_{li}, b_k]\right) + \mathbb{Cov}[b_k, b_l] \tag{43}$$

$$= \sum_{1 \leq i,j \leq D_{\text{in}}} \mathbb{Cov}[a_i W_{ki}, a_j W_{lj}] \tag{44}$$

$$= \sum_{1 \leq i,j \leq D_{\text{in}}} \mathbb{E}[a_i]\mathbb{E}[a_j]\mathbb{Cov}[W_{ki}, W_{lj}] + \mathbb{E}[W_{ki}]\mathbb{E}[W_{lj}]\mathbb{Cov}[a_i, a_j] + \mathbb{Cov}[a_i, a_j]\mathbb{Cov}[W_{ki}, W_{lj}]$$

$$\tag{45}$$

$$= \sum_{1 \leq i,j \leq D_{\text{in}}} \mathbb{E}[W_{ki}]\mathbb{E}[W_{lj}]\mathbb{Cov}[a_i, a_j]. \tag{46}$$

For $k = l$, we have $\mathbb{Var}[h_k] =$

$$\sum_{1 \leq i,j \leq D_{\text{in}}} \mathbb{Cov}[a_i W_{ki}, a_j W_{kj}] + \sum_{i=1}^{D_{\text{in}}} \left(\mathbb{E}[a_i]\mathbb{Cov}[W_{ki}, b_k] + \mathbb{E}[a_i]\mathbb{Cov}[W_{ki}, b_k]\right) + \mathbb{Var}[b_k] \tag{47}$$

$$= \sum_{1 \leq i \leq D_{\text{in}}} \mathbb{Cov}[a_i W_{ki}, a_i W_{ki}] + \mathbb{Var}[b_k] \tag{48}$$

$$= \sum_{1 \leq i \leq D_{\text{in}}} \mathbb{E}[a_i]^2 \mathbb{Var}[W_{ki}] + \mathbb{E}[W_{ki}]^2 \mathbb{Var}[a_i] + \mathbb{Var}[a_i]\mathbb{Var}[W_{ki}] + \mathbb{Var}[b_k]. \tag{49}$$

Note that as

$$\mathbb{V}\text{ar}\left[h_k^{(m)}\right] = \sum_{1 \le i \le D_{\text{in}}} \mathbb{E}\left[a_i^{(m-1)}\right]^2 \mathbb{V}\text{ar}\left[W_{ki}^{(m)}\right] + \mathbb{E}\left[W_{ki}^{(m)}\right]^2 \mathbb{V}\text{ar}\left[a_i^{(m-1)}\right] \tag{50}$$

$$+ \sum_{1 \le i \le D_{\text{in}}} \mathbb{V}\text{ar}\left[a_i^{(m-1)}\right] \mathbb{V}\text{ar}\left[W_{ki}^{(m)}\right] + \mathbb{V}\text{ar}\left[b_k^{(m)}\right], \tag{51}$$

the variance of $h_k^{(m)}$ will only rely on the variance of the activations of previous layers, *i.e.*, $\mathbb{V}\text{ar}[a_i^{(m-1)}]$. In the case of element-wise activation functions, $\mathbb{V}\text{ar}[a_i^{(m-1)}]$ will only rely on $\mathbb{V}\text{ar}[h_i^{(m-1)}]$ as now the Jacobian of the activation is diagonal. As a result, in the case where we only need the variance of the input, we could drop the computation of $\mathbb{C}\text{ov}[h_k, h_l]$ and only compute the variance for each layer, which will largely reduce the computation cost.

### A.4 Derivation for Kronecker Covariance Structure

In KFAC, the Hessian is represented in Kronecker product form $\mathbf{Hess} = \boldsymbol{A} \otimes \boldsymbol{B}$. Denote the prior precision as $\lambda^2$, then the posterior covariance is

$$\boldsymbol{\Sigma} = (\mathbf{Hess} + \lambda^2 \mathbf{I})^{-1} = (\boldsymbol{A} \otimes \boldsymbol{B} + \lambda^2 \mathbf{I})^{-1} \tag{52}$$

As there is no closed form for the inverse, to express the covariance in the form of the Kronecker product as well, we approximate the covariance as

$$\boldsymbol{\Sigma} = (\boldsymbol{A} \otimes \boldsymbol{B} + \lambda^2 \mathbf{I})^{-1} \tag{53}$$

$$= \left[(\boldsymbol{U}_A \boldsymbol{\Lambda}_A \boldsymbol{U}_A^\top) \otimes (\boldsymbol{U}_B \boldsymbol{\Lambda}_B \boldsymbol{U}_B^\top) + \lambda^2 \mathbf{I}\right]^{-1} \quad \text{(Eigen Decomposition)}$$

$$\approx \left[(\boldsymbol{U}_A(\boldsymbol{\Lambda}_A + \lambda \mathbf{I}_A)\boldsymbol{U}_A^\top) \otimes (\boldsymbol{U}_B(\boldsymbol{\Lambda}_B + \lambda \mathbf{I}_B)\boldsymbol{U}_B^\top)\right]^{-1} \tag{54}$$

$$= \underbrace{\left(\boldsymbol{U}_A(\boldsymbol{\Lambda}_A + \lambda \mathbf{I}_A)\boldsymbol{U}_A^\top\right)^{-1}}_{\boldsymbol{C}} \otimes \underbrace{\left(\boldsymbol{U}_B(\boldsymbol{\Lambda}_B + \lambda \mathbf{I}_B)\boldsymbol{U}_B^\top\right)^{-1}}_{\boldsymbol{D}}. \quad ((\boldsymbol{A} \otimes \boldsymbol{B})^{-1} = \boldsymbol{A}^{-1} \otimes \boldsymbol{B}^{-1})$$

Recall for an efficient implementation for computing $\sum_{1 \le i,j \le D_{\text{in}}} \mathbb{C}\text{ov}[a_i W_{ki}, a_j W_{lj}]$ (Eq. (35)), we need to retrieve the covariance between the $k^{\text{th}}$ row of weight and $l^{\text{th}}$ row of weight, which is a $D_{\text{in}} \times D_{\text{in}}$ matrix:

$$\mathbb{C}\text{ov}\left[\boldsymbol{W}[k,:], \boldsymbol{W}[l,:]\right] = \begin{bmatrix} \mathbb{C}\text{ov}[W_{k1}, W_{l1}] & \dots & \mathbb{C}\text{ov}[W_{k1}, W_{lD_{\text{in}}}] \\ \vdots & \ddots & \vdots \\ \mathbb{C}\text{ov}[W_{kD_{\text{in}}}, W_{l1}] & \dots & \mathbb{C}\text{ov}[W_{kD_{\text{in}}}, W_{lD_{\text{in}}}] \end{bmatrix}. \tag{55}$$

As $\boldsymbol{\Sigma} \approx \boldsymbol{C} \otimes \boldsymbol{D}$ where $\boldsymbol{C} \in \mathbb{D}_{\text{in}} \times \mathbb{D}_{\text{in}}$ and $\boldsymbol{D} \in \mathbb{D}_{\text{out}} \times \mathbb{D}_{\text{out}}$, the posterior covariance is represented by a total number of $D_{\text{in}} \times D_{\text{in}}$ matrix with size $D_{\text{out}} \times D_{\text{out}}$. Retrieving a $D_{\text{in}} \times D_{\text{in}}$ matrix from it is not trivial. In the toy example as shown in Fig. 7, for a $D_{\text{in}} = 3$ and $D_{\text{out}} = 2$ matrix $\boldsymbol{W}$, its covariance is represented by a total number of 9 ($D_{\text{in}} \times D_{\text{in}}$) matrix I, II, ..., IX with shape $2 \times 2$ ($D_{\text{out}} \times D_{\text{out}}$). To retrieve $\mathbb{C}\text{ov}[\boldsymbol{W}[1,:], \boldsymbol{W}[2,:]]$, we need to first decide which Kronecker blocks contain it (in this case block II, III, V and VI) and reconstruct these Kronecker blocks. Then, we retrieve $\mathbb{C}\text{ov}[\boldsymbol{W}[1,:], \boldsymbol{W}[2,:]]$ from the reconstructed blocks.

In general, the retrieval process consists of two steps: (1) identifying the block indices within the Kronecker product matrix that correspond to the required covariance block and (2) extracting the covariance of interests from the constructed block.

**Identifying Block Indices** We first identify the Kronecker blocks that contain the covariance of interest. This is achieved by calculating the block indexed for $\boldsymbol{C}$, which is later used to construct Kronecker blocks. Specifically, the start and end positions of the covariance block corresponding to rows $k$ and $l$ can be computed as:

$$\text{row\_start} = \left\lfloor \frac{k \cdot D_{\text{in}}}{D_{\text{out}}} \right\rfloor, \tag{56}$$

$$\mathbb{C}\text{ov}[\boldsymbol{W}] = \begin{pmatrix} \begin{array}{cc|cc|cc} W_{11}, W_{21} & W_{11}, W_{23} & W_{11}, W_{33} & W_{11}, W_{31} & W_{11}, W_{32} & W_{11}, W_{33} \\ W_{12}, W_{11} & W_{12}, W_{12} & W_{12}, W_{13} & W_{12}, W_{31} & W_{12}, W_{32} & W_{12}, W_{33} \\ \hline W_{13}, W_{11} & W_{13}, W_{13} & W_{13}, W_{13} & W_{13}, W_{31} & W_{13}, W_{33} & W_{13}, W_{33} \\ W_{21}, W_{11} & W_{21}, W_{12} & W_{21}, W_{13} & W_{21}, W_{31} & W_{21}, W_{32} & W_{21}, W_{33} \\ \hline W_{22}, W_{11} & W_{22}, W_{12} & W_{22}, W_{13} & W_{22}, W_{31} & W_{22}, W_{32} & W_{22}, W_{33} \\ W_{23}, W_{11} & W_{23}, W_{12} & W_{23}, W_{13} & W_{23}, W_{31} & W_{23}, W_{32} & W_{23}, W_{33} \end{array} \end{pmatrix}$$

Figure 7: To retrieve the highlighted submatrix $\mathbb{C}\text{ov}[\boldsymbol{W}[1,:], \boldsymbol{W}[2,:]]$ of the covariance for $\boldsymbol{W} \in \mathbb{R}^{2\times3}$, we identify the Kronecker blocks that contain the covariance of interest (II, III, V, and VI), explicate those blocks in memory, and then retrieve the relevant submatrix.

$$\text{row\_end} = \left\lceil \frac{(k+1) \cdot D_{\text{in}}}{D_{\text{out}}} \right\rceil, \tag{57}$$

$$\text{col\_start} = \left\lfloor \frac{l \cdot D_{\text{in}}}{D_{\text{out}}} \right\rfloor, \tag{58}$$

$$\text{col\_end} = \left\lceil \frac{(l+1) \cdot D_{\text{in}}}{D_{\text{out}}} \right\rceil. \tag{59}$$

Then, we can construct the Kronecker blocks that contain the covariance of interest by $\boldsymbol{C}[\text{row\_start} : \text{row\_end}, \text{col\_start} : \text{col\_end}]\boldsymbol{D}$.

**Extract the Covariance** Once we have $\boldsymbol{C}[\text{row\_start} : \text{row\_end}, \text{col\_start} : \text{col\_end}]$, and as we know the covariance we need to retrieve has shape $D_{\text{in}} \times D_{\text{in}}$, we only need to compute the start row and column index, which can be computed as

$$\text{select\_row\_start} = (k \cdot D_{\text{in}}) \bmod D_{\text{out}}, \tag{60}$$

$$\text{select\_col\_start} = (l \cdot D_{\text{in}}) \bmod D_{\text{out}}. \tag{61}$$

## A.5 DERIVATION FOR ACTIVATION LAYERS

For $\boldsymbol{a} = g(\boldsymbol{h})$ where $\boldsymbol{h} \sim \mathcal{N}(\boldsymbol{h}; \mathbb{E}[\boldsymbol{h}], \boldsymbol{\Sigma}_h)$ and $g(\cdot)$ is the activation function, we use local linearisation to approximate the distribution of $\boldsymbol{a}$. Specifically, we do a first-order Taylor expansion on $g(\cdot)$ at $\mathbb{E}[\boldsymbol{h}]$:

$$\boldsymbol{a} = g(\boldsymbol{h}) \tag{62}$$
$$\approx g(\mathbb{E}[\boldsymbol{h}]) + \boldsymbol{J}_g|_{\boldsymbol{h}=\mathbb{E}[\boldsymbol{h}]}(\boldsymbol{h} - \mathbb{E}[\boldsymbol{h}]). \tag{63}$$

Given that Gaussian distribution is closed under linear transformation, we have

$$\boldsymbol{h} \sim \mathcal{N}(\mathbb{E}[\boldsymbol{h}], \boldsymbol{\Sigma}_h) \tag{64}$$
$$\boldsymbol{h} - \mathbb{E}[\boldsymbol{h}] \sim \mathcal{N}(\boldsymbol{0}, \boldsymbol{\Sigma}_h) \tag{65}$$
$$\boldsymbol{J}_g|_{\boldsymbol{h}=\mathbb{E}[\boldsymbol{h}]}(\boldsymbol{h} - \mathbb{E}[\boldsymbol{h}]) \sim \mathcal{N}(\boldsymbol{0}, \boldsymbol{J}_g|_{\boldsymbol{h}=\mathbb{E}[\boldsymbol{h}]}^{\top} \boldsymbol{\Sigma}_h \boldsymbol{J}_g|_{\boldsymbol{h}=\mathbb{E}[\boldsymbol{h}]}) \tag{66}$$
$$g(\mathbb{E}[\boldsymbol{h}]) + \boldsymbol{J}_g|_{\boldsymbol{h}=\mathbb{E}[\boldsymbol{h}]}(\boldsymbol{h} - \mathbb{E}[\boldsymbol{h}]) \sim \mathcal{N}(g(\mathbb{E}[\boldsymbol{h}]), \boldsymbol{J}_g|_{\boldsymbol{h}=\mathbb{E}[\boldsymbol{h}]}^{\top} \boldsymbol{\Sigma}_h \boldsymbol{J}_g|_{\boldsymbol{h}=\mathbb{E}[\boldsymbol{h}]}) \tag{67}$$
$$\boldsymbol{a} \underset{\text{approx}}{\sim} \mathcal{N}(\boldsymbol{a}; g(\mathbb{E}[\boldsymbol{h}]), \boldsymbol{J}_g|_{\boldsymbol{h}=\mathbb{E}[\boldsymbol{h}]}^{\top} \boldsymbol{\Sigma}_h \boldsymbol{J}_g|_{\boldsymbol{h}=\mathbb{E}[\boldsymbol{h}]}). \tag{68}$$

## A.6 TRANSFORMER BLOCK

There are four components in each transformer block (Vaswani et al., 2017): (1) multi-head attention; (2) MLP; (3) layer normalisation; and (4) residual connection. For MLP blocks, the propagation is the same as described above. For layer normalisation and residual connection, as Gaussian distributions are closed under linear transformations, 'pushing' distributions through them is straightforward.

We describe how to push distributions through attention layers below. Note that for computational reasons, we always assume the input has diagonal covariance.

Given an input $\boldsymbol{H} \in \mathbb{R}^{T \times D}$ where $T$ is the number of tokens in the input sequence and $D$ is the dimension of each token, denote the query, key and value matrices as $\boldsymbol{W}_Q \in \mathbb{R}^{D \times D}$, $\boldsymbol{W}_K \in \mathbb{R}^{D \times D}$, $\boldsymbol{W}_V \in \mathbb{R}^{D \times D}$ respectively, the key, query and value in an attention blocks are

$$\boldsymbol{Q} = \boldsymbol{H}\boldsymbol{W}_Q, \quad \boldsymbol{K} = \boldsymbol{H}\boldsymbol{W}_K, \quad \boldsymbol{V} = \boldsymbol{H}\boldsymbol{W}_V, \tag{69}$$

and the output of attention block is

$$\text{Attention}(\boldsymbol{H}) = \text{Softmax}(\frac{\boldsymbol{Q}\boldsymbol{K}^\top}{\sqrt{D}})\boldsymbol{V}. \tag{70}$$

When the input $\boldsymbol{H}$ is a distribution, $\boldsymbol{Q}$, $\boldsymbol{K}$ and $\boldsymbol{V}$ will all be distributions as well. As pushing a distribution over a softmax activation requires further approximation, we ignore the distribution over $\boldsymbol{Q}$ and $\boldsymbol{K}$ for computational reasons and compute their value by using the mean of the input:

$$\boldsymbol{Q} = \mathbb{E}\left[\boldsymbol{H}\right] \mathbb{E}\left[\boldsymbol{W}_Q\right], \quad \boldsymbol{K} = \mathbb{E}\left[\boldsymbol{H}\right] \mathbb{E}\left[\boldsymbol{W}_K\right]. \tag{71}$$

For $\boldsymbol{V}$, for simplicity we describe our approximation for a single token $\boldsymbol{h}$ whose value is $\boldsymbol{v} = \boldsymbol{W}_V\boldsymbol{h}$ with $k^{\text{th}}$ element being $v_k = \sum_{i=1}^{D} W_{V_{ki}} h_i$. Assuming $\boldsymbol{h}$ is a Gaussian, the covariance between the $k^{\text{th}}$ and the $l^{\text{th}}$ value is

$$\mathbb{C}\text{ov}\left[v_k, v_l\right] = \mathbb{C}\text{ov}\left[\sum_{i=1}^{D} W_{V_{ki}} h_i, \sum_{j=1}^{D} W_{V_{lj}} h_j\right] \tag{72}$$

$$= \sum_{i=1}^{D}\sum_{j=1}^{D} \mathbb{C}\text{ov}\left[W_{V_{ki}} h_i, W_{V_{lj}} h_j\right]. \tag{73}$$

When treating $\boldsymbol{W}_V$ deterministically, we have

$$\mathbb{C}\text{ov}\left[v_k, v_l\right] = \sum_{i=1}^{D}\sum_{j=1}^{D} \mathbb{C}\text{ov}\left[W_{V_{ki}} h_i, W_{V_{lj}} h_j\right] \qquad \text{(definition)}$$

$$= \sum_{i=1}^{D}\sum_{j=1}^{D} W_{V_{ki}} W_{V_{lj}} \mathbb{C}\text{ov}\left[h_i, h_j\right] \qquad (\boldsymbol{W}_V \text{ deterministic})$$

$$\approx \sum_{i=1}^{D} W_{V_{ki}} W_{V_{li}} \mathbb{V}\text{ar}\left[h_i\right]. \quad \text{(ignore correlation between } \boldsymbol{h} \text{ for computational reason)}$$

When $\boldsymbol{W}_V$ is an isotropic Gaussian, we have

$$\mathbb{C}\text{ov}\left[v_k, v_l\right] = \sum_{1 \le i,j \le D} \mathbb{C}\text{ov}\left[W_{V_{ki}} h_i, W_{V_{lj}} h_j\right] \tag{74}$$

$$\approx \sum_{1 \le i,j \le D} \left(\mathbb{E}\left[h_i\right] \mathbb{E}\left[h_j\right] + \mathbb{C}\text{ov}\left[h_i, h_j\right]\right) \mathbb{C}\text{ov}\left[W_{ki}, W_{lj}\right] + \mathbb{E}\left[W_{ki}\right] \mathbb{E}\left[W_{lj}\right] \mathbb{C}\text{ov}\left[h_i, h_j\right]$$

$$\text{(assumption A2)}$$

$$= \sum_{1 \le i,j \le D} \mathbb{E}\left[W_{ki}\right] \mathbb{E}\left[W_{lj}\right] \mathbb{C}\text{ov}\left[h_i, h_j\right] \qquad (\boldsymbol{W}_V \text{ is isotropic Gaussian})$$

$$\approx \sum_{1 \le i \le D} \mathbb{E}\left[W_{ki}\right] \mathbb{E}\left[W_{li}\right] \mathbb{V}\text{ar}\left[h_i\right].$$

$$\text{(ignore correlation between } \boldsymbol{h} \text{ for computational reason)}$$

$$
\begin{aligned}
\mathbb{V}\text{ar}\left[v_k\right] &= \sum_{1 \le i,j \le D} \mathbb{C}\text{ov}\left[W_{V_{ki}} h_i, W_{V_{kj}} h_j\right] && \text{(definition)} \\
&\approx \sum_{1 \le i,j \le D} \left(\mathbb{E}\left[h_i\right] \mathbb{E}\left[h_j\right] + \mathbb{C}\text{ov}\left[h_i, h_j\right]\right) \mathbb{C}\text{ov}\left[W_{ki}, W_{kj}\right] + \mathbb{E}\left[W_{ki}\right] \mathbb{E}\left[W_{kj}\right] \mathbb{C}\text{ov}\left[h_i, h_j\right] \\
&&& \text{(assumption A2)} \\
&= \sum_{1 \le i \le D} \left(\mathbb{E}\left[h_i\right]^2 + \mathbb{V}\text{ar}\left[h_i\right]\right) \mathbb{V}\text{ar}\left[W_{ki}\right] + \mathbb{E}\left[W_{ki}\right]^2 \mathbb{V}\text{ar}\left[h_i\right]. \\
&&& (\boldsymbol{W}_V \text{ is isotropic Gaussian})
\end{aligned}
\tag{75}
$$

Once we have the distribution over $\boldsymbol{V}$, the distribution over Attention($\boldsymbol{H}$) becomes a distribution of linear combination of Gaussian, which is tractable.

Then for multi-head attention, we assume each attention head's output is independent, which allows us to compute the distribution over the final output in tractable form. As we assume the input is isotropic, we only need to compute the variance for each dimension.

### A.7 CONVOLUTIONAL NEURAL NETWORK

The derivation for convolutional layers is similar to fully connected layers as convolution layers can be considered as a shared weight fully connected layer. We first give the derivation for convolutional layers, then discuss pooling layers in convolutional neural networks.

Denote the pixel value at $(i, j)$ of $c_{\text{in}}{}^{\text{th}}$ channel as $a_{c_{\text{in}}}[i, j]$, the $c_{\text{in}}{}^{\text{th}}$ channel of convolutional kernel corresponding to $c_{\text{out}}{}^{\text{th}}$ output channel as $W_{c_{\text{out}}, c_{\text{in}}}[i, j]$ and the pixel value at $(k, l)$ of the $c_{\text{out}}{}^{\text{th}}$ output channel as $h_{c_{\text{out}}}[k, l]$. Then, suppose there are $C_{\text{in}}$ channels in total and the kernel size is $K_h \times K_w$, we can write the convolutional layer as

$$
h_{c_{\text{out}}}[k, l] = \sum_{c_{\text{in}}=1}^{C_{\text{in}}} \sum_{i=1}^{K_h} \sum_{j=1}^{K_w} a_{c_{\text{in}}}[k + i - 1, l + j - 1] W_{c_{\text{out}}, c_{\text{in}}}[i, j].
\tag{76}
$$

**Derivation of mean** Following our assumption that $a_{c_{\text{in}}}[k + i - 1, l + j - 1]$ is uncorrelated with $W_{c_{\text{out}}, c_{\text{in}}}[i, j]$, we have

$$
\mathbb{E}\left[h_{c_{\text{out}}}[k, l]\right] = \sum_{c_{\text{in}}=1}^{C_{\text{in}}} \sum_{i=1}^{K_h} \sum_{j=1}^{K_w} \mathbb{E}\left[a_{c_{\text{in}}}[k + i - 1, l + j - 1]\right] \mathbb{E}\left[W_{c_{\text{out}}, c_{\text{in}}}[i, j]\right]
\tag{77}
$$

**Derivation of covariance** The covariance between pixels of the $c_{\text{out}}{}^{\text{th}}$ output channel are given as:

$$
\mathbb{C}\text{ov}\left[h_{c_{\text{out}}}[k_1, l_1], h_{c_{\text{out}}}[k_2, l_2]\right]
\tag{78}
$$

$$
= \mathbb{C}\text{ov}\left[ \sum_{c_{\text{in},1}=1}^{C_{\text{in}}} \sum_{i_1=1}^{K_h} \sum_{j_1=1}^{K_w} a_{c_{\text{in},1}}[k_1 + i_1 - 1, l_1 + j_1 - 1] W_{c_{\text{out}}, c_{\text{in}}}[i_1, j_1] , \right.
\tag{79}
$$

$$
\left. \sum_{c_{\text{in},2}=1}^{C_{\text{in}}} \sum_{i_2=1}^{K_h} \sum_{j_2=1}^{K_w} a_{c_{\text{in},2}}[k_2 + i_2 - 1, l_2 + j_2 - 1] W_{c_{\text{out}}, c_{\text{in}}}[i_2, j_2] \right]
\tag{80}
$$

$$
= \sum_{c_{\text{in},1}=1}^{C_{\text{in}}} \sum_{i_1=1}^{K_h} \sum_{j_1=1}^{K_w} \sum_{c_{\text{in},2}=1}^{C_{\text{in}}} \sum_{i_2=1}^{K_h} \sum_{j_2=1}^{K_w} \mathbb{C}\text{ov}\left[a_{c_{\text{in},1}}[k_1 + i_1 - 1, l_1 + j_1 - 1] W_{c_{\text{out}}, c_{\text{in}}}[i_1, j_1] \right.
\tag{81}
$$

$$
\left. a_{c_{\text{in},2}}[k_2 + i_2 - 1, l_2 + j_2 - 1] W_{c_{\text{out}}, c_{\text{in}}}[i_2, j_2] \right].
\tag{82}
$$

Using earlier results from Eq. (22) and the shorthand $W = W c_{\text{out}}, c_{\text{in}}$, we have:

$$\mathbb{C}\text{ov}\left[a_{\text{in},1}[k_1+i_1-1, l_1+j_1-1]W[i_1,j_1], a_{\text{in},2}[k_2+i_2-1, l_2+j_2-1]W[i_2,j_2]\right] \tag{83}$$

$$\approx \mathbb{E}\left[a_{\text{in},1}[k_1+i_1-1, l_1+j_1-1]a_{\text{in},2}[k_2+i_2-1, l_2+j_2-1]\right]\mathbb{E}\left[W[i_1,j_1]W[i_2,j_2]\right] \tag{84}$$

$$- \mathbb{E}\left[a_{\text{in},1}[k_1+i_1-1, l_1+j_1-1]\right]\mathbb{E}\left[a_{\text{in},2}[k_2+i_2-1, l_2+j_2-1]\right]\mathbb{E}\left[W[i_1,j_1]\right]\mathbb{E}\left[W[i_2,j_2]\right] \tag{85}$$

$$= \mathbb{E}\left[a_{\text{in},1}[k_1+i_1-1, l_1+j_1-1]\right]\mathbb{E}\left[a_{\text{in},2}[k_2+i_2-1, l_2+j_2-1]\right]\mathbb{C}\text{ov}\left[W[i_1,j_1], W[i_2,j_2]\right] \tag{86}$$

$$+ \mathbb{E}\left[W[i_1,j_1]\right]\mathbb{E}\left[W[i_2,j_2]\right]\mathbb{C}\text{ov}\left[a_{\text{in},1}[k_1+i_1-1, l_1+j_1-1], a_{\text{in},2}[k_2+i_2-1, l_2+j_2-1]\right] \tag{87}$$

$$+ \mathbb{C}\text{ov}\left[a_{\text{in},1}[k_1+i_1-1, l_1+j_1-1], a_{\text{in},2}[k_2+i_2-1, l_2+j_2-1]\right]\mathbb{C}\text{ov}\left[W[i_1,j_1], W[i_2,j_2]\right]. \tag{88}$$

## B  ADDITIONAL EXPERIMENTS

In the Appendix, we provide additional details on (i) the regression experiments App. B.1, (ii) the classification experiments App. B.2, and (iii) the image sensitivity experiment App. B.3. In addition, we also present additional experiments on (i) measuring the performance by varying the number of MC samples for the sampling baseline App. B.4, (ii) estimating the degree of local linearity in our method App. B.5, and (iii) comparing the runtime of our method against the baselines App. B.7 to further demonstrate the benefits of our streamlined prediction with local linearisation and local Gaussian approximation.

### B.1  REGRESSION

Table 7 gives the UCI regression data set information and the neural network structure we used. For all neural networks, we use the ReLU activation function. In Table 8, we report the Root Mean Square Error (RMSE). Our method results in matching or better performance compared with sampling and GLM, indicating the effectiveness of our method. Note that as the mean of the posterior prediction of our method is the same as the prediction made by setting the weights of the neural network to be the mean of the posterior, we result in the same prediction as GLM of LA, and hence the same performance.

Table 7: UCI regression experiment setup.

| Data Set Name | Shorthand | $(n, d)$ | Network Structure |
|---|---|---|---|
| SERVO | SERVO | (167, 4) | $d$-50-1 |
| LIVER DISORDERS | LD | (345, 5) | $d$-50-1 |
| AUTO MPG | AM | (398, 7) | $d$-50-1 |
| REAL ESTATE VALUATION | REV | (414,6) | $d$-50-1 |
| FOREST FIRES | FF | (517, 12) | $d$-50-1 |
| INFRARED THERMOGRAPHY TEMPERATURE | ITT | (1020, 33) | $d$-100-1 |
| CONCRETE COMPRESSIVE STRENGTH | CCS | (1030, 8) | $d$-100-1 |
| AIRFOIL SELF-NOISE | ASN | (1503, 5) | $d$-100-1 |
| COMMUNITIES AND CRIME | CAC | (1994, 127) | $d$-100-1 |
| PARKINSONS TELEMONITORING | PT | (5875, 19) | $d$-50-50-1 |
| COMBINED CYCLE POWER PLANT | CCPP | (9568, 4) | $d$-50-50-1 |

### B.2  CLASSIFICATION

Table 9 gives the classification data sets information and the neural network structure we used for the MLP experiment. We use ReLU activation for MLP.

**OOD Experiments with MLP**  To test our method on out-of-distribution (OOD) data, we first evaluate the MNIST-trained MLP on rotated versions of the test set as shown in Fig. 8. The rotation degree interval is $10°$ from $0 - 180°$. We observe that with increasing rotation degree, our method achieves a lower NLPD compared to LA MAP and MFVI Sampling while being close compared with LA Sampling and GLM. Also, our method achieves similar NLPD for both LA and MFVI posterior approximations across the rotation degrees. All methods perform on par regarding their ACC. In Fig. 9, we show kernel density plots over the predictive entropy of an FMNIST-trained MLP evaluated on MNIST. Our method can distinguish between in-distribution and OOD data better than the LA MAP and MFVI Sampling. Although our method under fits the in-distribution data, the separation between them is clear for the OOD data.

Table 8: Root Mean Square Error ↓ on UCI regression data sets. Our method results in better or matching performance compared with sampling and GLM, indicating its effectiveness.

| | | MFVI (Diag. Cov.) | | Laplace Approximation (Full Cov.) | | |
|---|---|---|---|---|---|---|
| | $(n, d)$ | Sampling | Ours | Sampling | GLM | Ours |
| SERVO | (167, 4) | $0.749_{\pm 0.147}$ | $\mathbf{0.740}_{\pm 0.143}$ | $1.632_{\pm 0.233}$ | $\mathbf{0.658}_{\pm 0.141}$ | $\mathbf{0.658}_{\pm 0.141}$ |
| LD | (345, 5) | $\mathbf{0.884}_{\pm 0.273}$ | $\mathbf{0.881}_{\pm 0.272}$ | $0.989_{\pm 0.441}$ | $\mathbf{0.977}_{\pm 0.418}$ | $\mathbf{0.977}_{\pm 0.418}$ |
| AM | (398, 7) | $\mathbf{0.415}_{\pm 0.115}$ | $0.417_{\pm 0.113}$ | $0.505_{\pm 0.105}$ | $\mathbf{0.371}_{\pm 0.103}$ | $\mathbf{0.371}_{\pm 0.103}$ |
| REV | (414, 6) | $\mathbf{0.563}_{\pm 0.096}$ | $\mathbf{0.562}_{\pm 0.095}$ | $0.789_{\pm 0.130}$ | $\mathbf{0.532}_{\pm 0.104}$ | $\mathbf{0.532}_{\pm 0.104}$ |
| FF | (517, 12) | $\mathbf{0.874}_{\pm 1.123}$ | $\mathbf{0.874}_{\pm 1.124}$ | $\mathbf{0.910}_{\pm 0.824}$ | $\mathbf{0.852}_{\pm 0.792}$ | $\mathbf{0.852}_{\pm 0.792}$ |
| ITT | (1020, 33) | $\mathbf{0.481}_{\pm 0.057}$ | $0.497_{\pm 0.066}$ | $0.560_{\pm 0.075}$ | $\mathbf{0.507}_{\pm 0.072}$ | $\mathbf{0.507}_{\pm 0.072}$ |
| CCS | (1030, 8) | $\mathbf{0.472}_{\pm 0.102}$ | $0.476_{\pm 0.106}$ | $0.494_{\pm 0.102}$ | $\mathbf{0.301}_{\pm 0.057}$ | $\mathbf{0.301}_{\pm 0.057}$ |
| ASN | (1503, 5) | $0.568_{\pm 0.062}$ | $\mathbf{0.560}_{\pm 0.062}$ | $0.550_{\pm 0.069}$ | $\mathbf{0.352}_{\pm 0.055}$ | $\mathbf{0.352}_{\pm 0.055}$ |
| CAC | (1994, 127) | $\mathbf{0.571}_{\pm 0.105}$ | $0.585_{\pm 0.092}$ | $1.481_{\pm 0.167}$ | $\mathbf{0.703}_{\pm 0.101}$ | $\mathbf{0.703}_{\pm 0.101}$ |
| PT | (5875, 19) | $0.601_{\pm 0.067}$ | $\mathbf{0.590}_{\pm 0.068}$ | $0.479_{\pm 0.081}$ | $\mathbf{0.410}_{\pm 0.076}$ | $\mathbf{0.410}_{\pm 0.076}$ |
| CCPP | (9568, 4) | $\mathbf{0.241}_{\pm 0.038}$ | $\mathbf{0.241}_{\pm 0.038}$ | $0.358_{\pm 0.041}$ | $\mathbf{0.224}_{\pm 0.037}$ | $\mathbf{0.224}_{\pm 0.037}$ |
| Bold Count | | 8/11 | 10/11 | 2/11 | 11/11 | 11/11 |

Table 9: Classification experiment setup.

| Data Set Name | $(n, d)$ | Network Structure |
|---|---|---|
| MNIST | (50000, 784) | $d$-128-64-10 |
| FMNIST | (50000, 784) | $d$-128-64-10 |
| ORGANCMNIST | (12975, 784) | $d$-128-64-11 |
| ORGANSMNIST | (13932, 784) | $d$-128-64-11 |

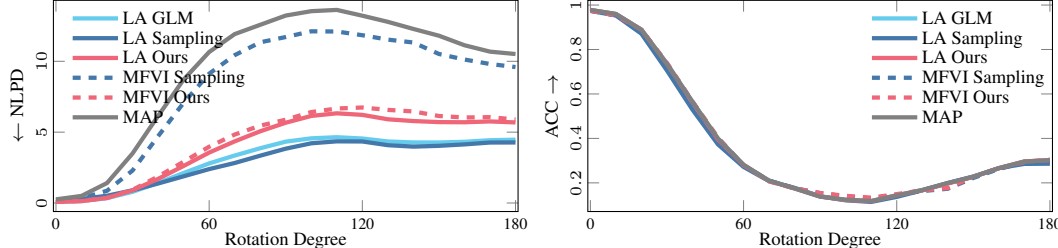

Figure 8: NLPD and ACC for MNIST-trained MLP on rotated versions of the MNIST test set. The rotation degree interval is $10°$ from $0 - 180°$. Our method achieves similar NLPD for both LA and MFVI posterior approximations.

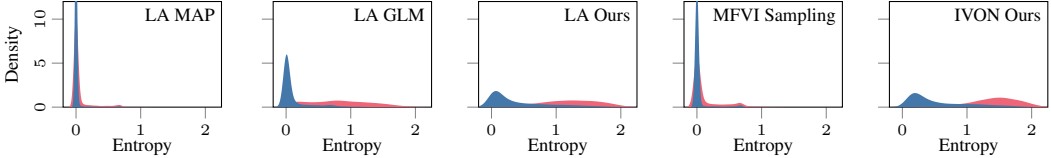

Figure 9: Kernel density plots over the predictive entropy from an MLP trained on FMNIST (blue, in-distribution) and data from MNIST (red, out-of-distribution). Our method results in a clear separation between the in- and out-of-distribution data.

**Our method applied to MLP in ViT** In Table 10 we report the results for fine-tuning the MLPs after the attention layers in the last two transformer blocks in ViT and later treating them Bayesian. We observe that our method achieves better or on par NLPD and ECE compared to the baselines for both LA and MFVI across all data sets while maintaining similar ACC as the baselines.

We present results for GPT-2 on tasks from GLUE (Wang et al., 2019b) and SuperGLUE (Wang et al., 2019a) benchmarks. These natural language understanding tasks could be turned into classification tasks with the prompt shown in Table 11. We add a classification layer on top of the encoder and use the embedding of the last token in each input to do classification.

Table 10: Performance metrics using ViT with posterior approximation on MLPs after the attention layers with the standard error for ACC and NLPD. Our method achieves better NLPD and ECE in general and achieves similar ACC compared to the baselines.

| Metrics | Methods | CIFAR-10 | CIFAR-100 | DTD | RESISC | IMAGENET-R |
|---|---|---|---|---|---|---|
| ACC ↑ | LA Sampling | $0.971_{\pm0.002}$ | $0.855_{\pm0.004}$ | $0.656_{\pm0.011}$ | $0.812_{\pm0.005}$ | $0.589_{\pm0.013}$ |
| | LA GLM | $\mathbf{0.974}_{\pm0.002}$ | $0.873_{\pm0.003}$ | $\mathbf{0.714}_{\pm0.010}$ | $0.886_{\pm0.004}$ | $\mathbf{0.687}_{\pm0.012}$ |
| | LA Ours | $\mathbf{0.976}_{\pm0.002}$ | $\mathbf{0.884}_{\pm0.003}$ | $\mathbf{0.716}_{\pm0.010}$ | $\mathbf{0.908}_{\pm0.004}$ | $\mathbf{0.713}_{\pm0.012}$ |
| | MFVI Sampling | $\mathbf{0.978}_{\pm0.001}$ | $\mathbf{0.896}_{\pm0.003}$ | $\mathbf{0.727}_{\pm0.010}$ | $\mathbf{0.870}_{\pm0.004}$ | $\mathbf{0.733}_{\pm0.012}$ |
| | MFVI Ours | $\mathbf{0.978}_{\pm0.001}$ | $\mathbf{0.895}_{\pm0.003}$ | $0.720_{\pm0.010}$ | $0.867_{\pm0.004}$ | $0.732_{\pm0.012}$ |
| NLPD ↓ | LA Sampling | $0.169_{\pm0.004}$ | $1.043_{\pm0.010}$ | $2.035_{\pm0.022}$ | $1.304_{\pm0.011}$ | $2.330_{\pm0.041}$ |
| | LA GLM | $0.089_{\pm0.005}$ | $0.602_{\pm0.011}$ | $1.260_{\pm0.029}$ | $0.568_{\pm0.011}$ | $1.584_{\pm0.045}$ |
| | LA Ours | $\mathbf{0.088}_{\pm0.006}$ | $\mathbf{0.457}_{\pm0.013}$ | $\mathbf{1.078}_{\pm0.036}$ | $\mathbf{0.318}_{\pm0.012}$ | $\mathbf{1.339}_{\pm0.047}$ |
| | MFVI Sampling | $0.124_{\pm0.011}$ | $0.480_{\pm0.018}$ | $\mathbf{1.277}_{\pm0.060}$ | $1.098_{\pm0.043}$ | $1.489_{\pm0.081}$ |
| | MFVI Ours | $\mathbf{0.081}_{\pm0.006}$ | $\mathbf{0.436}_{\pm0.013}$ | $1.146_{\pm0.040}$ | $0.650_{\pm0.019}$ | $\mathbf{1.206}_{\pm0.053}$ |
| ECE ↓ | LA Sampling | 0.078 | 0.349 | 0.442 | 0.431 | 0.331 |
| | LA GLM | $\mathbf{0.005}$ | 0.097 | 0.174 | 0.157 | 0.155 |
| | LA Ours | 0.007 | $\mathbf{0.032}$ | $\mathbf{0.055}$ | $\mathbf{0.017}$ | $\mathbf{0.087}$ |
| | MFVI Sampling | 0.014 | 0.040 | 0.083 | 0.075 | 0.115 |
| | MFVI Ours | $\mathbf{0.006}$ | $\mathbf{0.030}$ | $\mathbf{0.053}$ | $\mathbf{0.028}$ | $\mathbf{0.040}$ |

Table 11: Prompt templates for fine-tuning GPT-2 on natural language understanding tasks.

| Task | Prompt |
|---|---|
| MRPC | Answer whether sentence 2 is equivalent to sentence 1. Sentence 1: {sentence1}. Sentence 2: {sentence2}. Answer: |
| WiC | Select whether word {word} has the same meaning in these two sentences. Sentence 1: {sentence1}. Sentence 2: {sentence2}. Answer: |
| BoolQ | Answer the question with only True or False. Passage: {passage}. Question: {question}. Answer: |

**Lasy Layer Laplace Approximation on ViT**   In Table 12 we report the results for fine-tuning only the last classification layer in ViT base and later treating it Bayesian. We observe that our method (LL-LA Ours) achieves better or on par NLPD and ECE compared to last layer Laplace approximation (LL-LA GLM/Sampling) across all data sets while maintaining similar ACC. Compared to the case where more layers are treated Bayesian (LA Ours) (results are taken from Table 4), last layer approximations in general have lower accuracies and higher NLPD and ECE, *indicating the benefits gained by treating more layers Bayesian.* In Table 13 we report the wall-clock run times for last layer Laplace approximation on CIFAR-10 in milliseconds (see App. B.7 for the run time setting) Our method has matching speed with MAP and slight speed improvements over GLM.

Table 12: Performance metrics using ViT with posterior approximation on last layer with the standard error for ACC and NLPD. In the last layer Laplace approximation (LL-LA), our method achieves better NLPD and ECE in general and achieves similar ACC compared to the baselines. Compared with the case where more intermediate layers are treated Bayesian (LA Ours), last layer Laplace approximation in general has lower accuracies and higher NLPD and ECE.

| Metrics | Methods | CIFAR-10 | CIFAR-100 | DTD | RESISC | IMAGENET-R |
|---|---|---|---|---|---|---|
| ACC ↑ | LL-LA GLM | $\mathbf{0.965}_{\pm0.002}$ | $\mathbf{0.825}_{\pm0.004}$ | $\mathbf{0.681}_{\pm0.006}$ | $0.506_{\pm0.012}$ | $\mathbf{0.592}_{\pm0.013}$ |
| | LL-LA Sampling | $\mathbf{0.966}_{\pm0.002}$ | $\mathbf{0.827}_{\pm0.004}$ | $0.025_{\pm0.002}$ | $\mathbf{0.509}_{\pm0.012}$ | $\mathbf{0.604}_{\pm0.013}$ |
| | LL-LA Ours | $\mathbf{0.965}_{\pm0.002}$ | $\mathbf{0.825}_{\pm0.004}$ | $\mathbf{0.693}_{\pm0.006}$ | $\mathbf{0.508}_{\pm0.012}$ | $\mathbf{0.592}_{\pm0.013}$ |
| | LA Ours | $0.976_{\pm0.002}$ | $0.880_{\pm0.003}$ | $0.719_{\pm0.010}$ | $0.892_{\pm0.004}$ | $0.739_{\pm0.012}$ |
| NLPD ↓ | LL-LA GLM | $0.115_{\pm0.005}$ | $0.889_{\pm0.018}$ | $1.500_{\pm0.021}$ | $2.574_{\pm0.026}$ | $2.034_{\pm0.051}$ |
| | LL-LA Sampling | $0.118_{\pm0.005}$ | $0.924_{\pm0.021}$ | $7.341_{\pm0.065}$ | $2.456_{\pm0.030}$ | $2.000_{\pm0.058}$ |
| | LL-LA Ours | $\mathbf{0.110}_{\pm0.005}$ | $\mathbf{0.874}_{\pm0.019}$ | $\mathbf{1.411}_{\pm0.026}$ | $2.319_{\pm0.032}$ | $2.020_{\pm0.052}$ |
| | LA Ours | $0.086_{\pm0.006}$ | $0.456_{\pm0.012}$ | $1.068_{\pm0.035}$ | $0.353_{\pm0.012}$ | $1.264_{\pm0.043}$ |
| ECE ↓ | LL-LA GLM | 0.011 | 0.062 | 0.137 | 0.333 | 0.145 |
| | LL-LA Sampling | 0.015 | 0.045 | 0.215 | 0.304 | $\mathbf{0.127}$ |
| | LL-LA Ours | $\mathbf{0.007}$ | $\mathbf{0.029}$ | $\mathbf{0.026}$ | $\mathbf{0.233}$ | 0.135 |
| | LA Ours | 0.009 | 0.027 | 0.042 | 0.017 | 0.130 |

Table 13: Wallclock times for Last layer ViT base on CIFAR-10 in milliseconds.

| Model | Methods | Avg. Runtime ($\pm$ std) $\downarrow$ |
|---|---|---|
| Last Layer ViT | MAP | $3.732_{\pm 0.091}$ |
| | LA Sampling | $188.732_{\pm 0.051}$ |
| | LA GLM | $5.517_{\pm 0.033}$ |
| | Ours | $3.782_{\pm 0.088}$ |

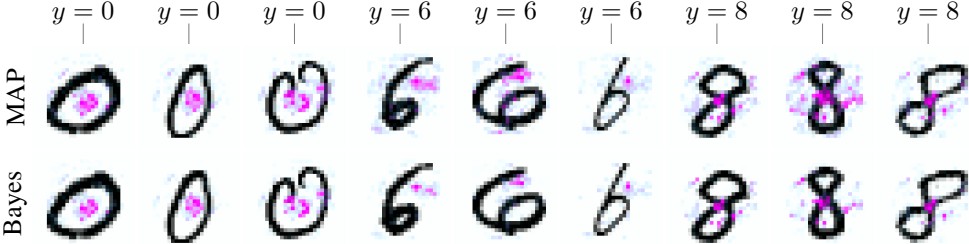

Figure 10: Pixel sensitivity of MLP classifiers trained on binary classification tasks ($0/6/8$) for MNIST digits. The rows show the sensitivity of the MAP predictor to pixel perturbations, and the pixel sensitivity for a last-layer Laplace approximation. The predictive distribution is approximated analytically in both cases. We observe that the Bayesian model using a Laplace approximation has less spurious sensitivities to pixel perturbations indicating that it is more robust to input perturbations. The sensitivities are visualised in the range ($0.5$ ▬▬ $1.0$)

### B.3 Image Pixel Sensitivity

We trained a $4$ layer MLP classifier on MNIST digits zero and eight using a batch size of $64$, learning rate of $1e-3$, weight decay set to $1e-5$, and for $50$ epochs. We used a subset of $0.1\%$ of the training data as held-out validation set and assumed a full covariance Gaussian distribution for each input centred at the pixel values of the datum and with a fixed covariance of $1e-5$. Furthermore, we then computed the pixel sensitivities for the trained model by learning the pixel-wise input covariance matrices by minimizing the negative log-likelihood of the held-out validation set and jointly maximizing the entropy of the input distributions. The optimisation was performed for each image independently and using Adam with a learning rate of $5e-3$ until the validation loss dropped below a divergence to the initial loss of $1e-2$. Doing so typically took around $700$ iterations. Fig. 10 shows some additional examples with the input-dependent sensitivities.

### B.4 Effect of the Number of MC Samples on Performance

We investigate the influence of number of samples on performance.

On regression tasks with small scale neural network (two layer MLP), we run experiments with the range of $[100, 500, 1000, 5000, 10000, 50000]$. On classification tasks with medium scale neural network (four layer MLP), we run experiments with the range of $[100, 500, 1000, 5000, 10000, 25000]$. On classification tasks with large scale neural network (ViT-Base), we run experiments with the range of $[10, 20, \ldots, 100]$. The results are reported in Figs. 11 to 13. The number of samples we used to report results in the main paper is shown in the dashed line.

In Fig. 11 and Fig. 13, we observe that for regression on small scale networks and classification on large scale networks, the performance saturates when set the number of samples to $50$ samples (classification with ViT-Base) or $1000$ samples (regression with two layer MLP). In Fig. 12, the performance saturates with $1000$ samples with LA. For MFVI the performance saturates with $5000$ samples, as the improvement gained on NLPD is marginal (from $2.13$ to $2.11$) from $1000$ samples to $5000$ samples, in experiment we set the number of samples as $1000$ for MFVI as well.

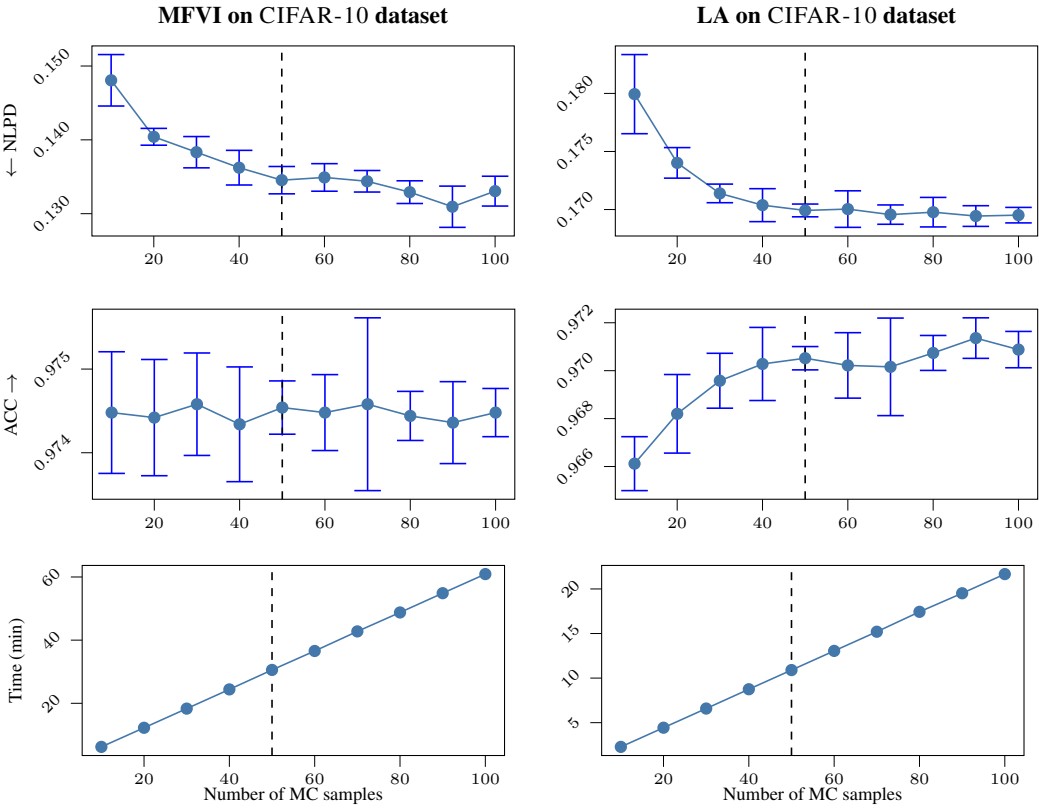

Figure 11: Effects of the number of MC samples on performance for LA and MFVI on CIFAR-10 with ViT-Base model. In the results reported in main paper, we set the number of MC samples to 50 (dashed line).

### B.5 ESTIMATING DEGREE OF LOCAL LINEARITY

We performed an additional experiment to assess the degree of local linearity of a trained MLP with ReLU activation functions. In particular, for trained MLP $f(\cdot)$, we are estimating the expected absolute error

$$\delta_{\text{Lin}} = \mathbb{E}_{\boldsymbol{z} \sim p(\boldsymbol{z})} \left[ |f(\boldsymbol{z}(1 \pm \epsilon)) - f(\boldsymbol{z})(1 \pm \epsilon)| \right], \tag{89}$$

where $\epsilon \geq 0$ and $\delta_{\text{Lin}}$ is zero for any $\epsilon$ if $f(\cdot)$ is linear around each $\boldsymbol{z}$.

In our experiments we vary $\epsilon$ in the range of $\epsilon \in [1\mathrm{e}{-6}, 1\mathrm{e}{-5}, \dots, 1]$ for a fully connected ReLU MLP with layers with sizes $[784, 128, 64, 10]$ trained on MNIST digits. After training, we removed the softmax operation on the last layer and measured the local linearisation error on the logits. We estimated the error on a random subset of $124$ validation data points and estimated the range of the inputs and the function outputs on the same subset. The range of input values is $3.246$ and the range of the function outputs varies between $153.072$ and $291.168$. Fig. 14 shows the results for each of the ten output dimensions scaled relative to their respective range. We observe that the trained ReLU MLP obtains low expected absolute error and behaves locally linear to a certain degree.

### B.6 COMPARISON OF VALUE COVARIANCE IN TRANSFORMERS

One way to improve efficiency in transformer is dropping the correlation between values, *i.e.*, drop the correlation $\mathbb{C}\text{ov}[v_k, v_l]$ given in Eq. (74). We compare the performance of both approximations and the results are given in Tables 14 to 16. Both approximation results in almost the same results for NLPD, ACC and ECE.

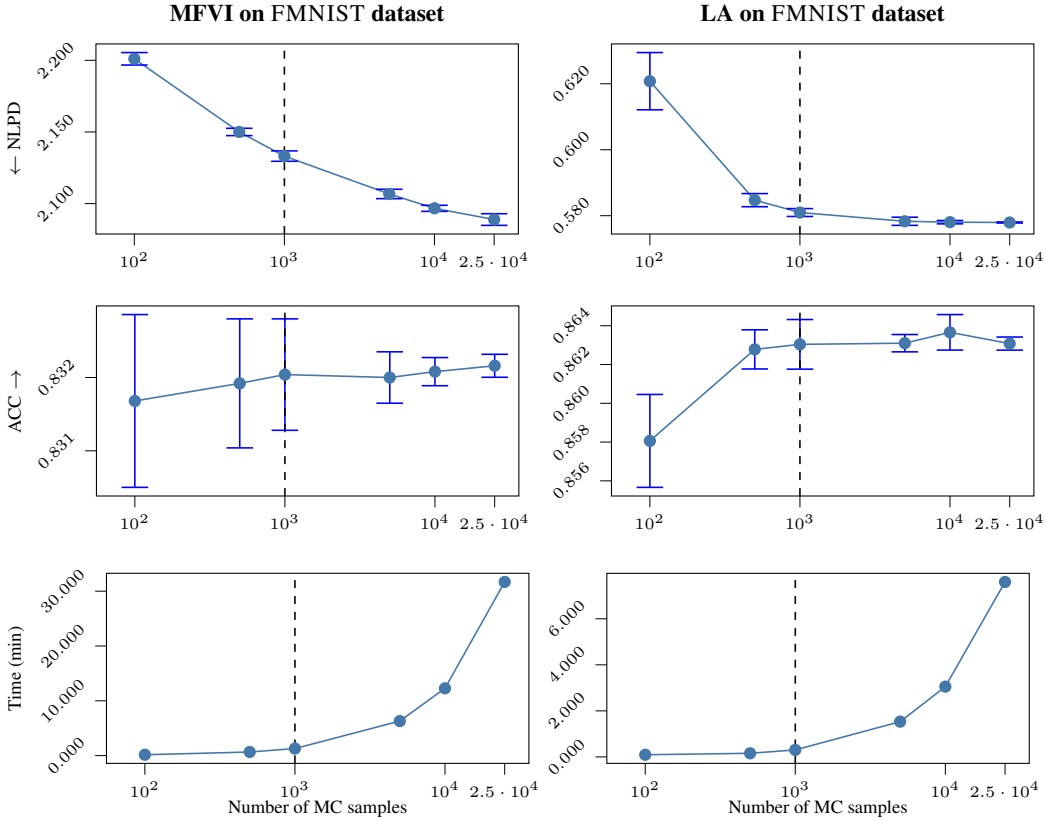

Figure 12: Effects of the number of MC samples on performance for LA and MFVI on FMNIST. In the results reported in main paper, we set the number of MC samples to 1000 (dashed line).

Table 14: Negative Log Predictive Density (NLPD) for ViT with posterior approximation in the attention layers. We compare only considering variance for value $V$ and considering full covariance. For MFVI and LA, both approximations results in almost the same result.

| Dataset | Mean Field Variational Inference | | Laplace Approximation | |
|---|---|---|---|---|
| | Full Covariance | Only Variance | Full Covariance | Only Variance |
| CIFAR-10 | $0.087 \pm 0.006$ | $0.088 \pm 0.006$ | $0.086 \pm 0.006$ | $0.086 \pm 0.006$ |
| CIFAR-100 | $0.468 \pm 0.012$ | $0.467 \pm 0.012$ | $0.456 \pm 0.012$ | $0.456 \pm 0.012$ |
| DTD | $1.006 \pm 0.035$ | $1.007 \pm 0.035$ | $1.068 \pm 0.035$ | $1.068 \pm 0.035$ |
| RESISC | $0.619 \pm 0.019$ | $0.616 \pm 0.019$ | $0.353 \pm 0.012$ | $0.352 \pm 0.012$ |
| IMAGENET-R | $1.234 \pm 0.052$ | $1.233 \pm 0.052$ | $1.264 \pm 0.043$ | $1.267 \pm 0.043$ |

Table 15: Accuracy (ACC) for ViT with posterior approximation n the attention layers. We compare only considering variance for value $V$ and considering full covariance. For MFVI and LA, both approximations results in almost the same result.

| Dataset | Mean Field Variational Inference | | Laplace Approximation | |
|---|---|---|---|---|
| | Full Covariance | Only Variance | Full Covariance | Only Variance |
| CIFAR-10 | $0.975 \pm 0.002$ | $0.975 \pm 0.004$ | $0.976 \pm 0.002$ | $0.976 \pm 0.004$ |
| CIFAR-100 | $0.880 \pm 0.003$ | $0.880 \pm 0.009$ | $0.880 \pm 0.003$ | $0.880 \pm 0.009$ |
| DTD | $0.734 \pm 0.010$ | $0.734 \pm 0.012$ | $0.719 \pm 0.010$ | $0.719 \pm 0.012$ |
| RESISC | $0.867 \pm 0.004$ | $0.867 \pm 0.009$ | $0.892 \pm 0.004$ | $0.892 \pm 0.008$ |
| IMAGENET-R | $0.727 \pm 0.012$ | $0.728 \pm 0.012$ | $0.739 \pm 0.012$ | $0.739 \pm 0.012$ |

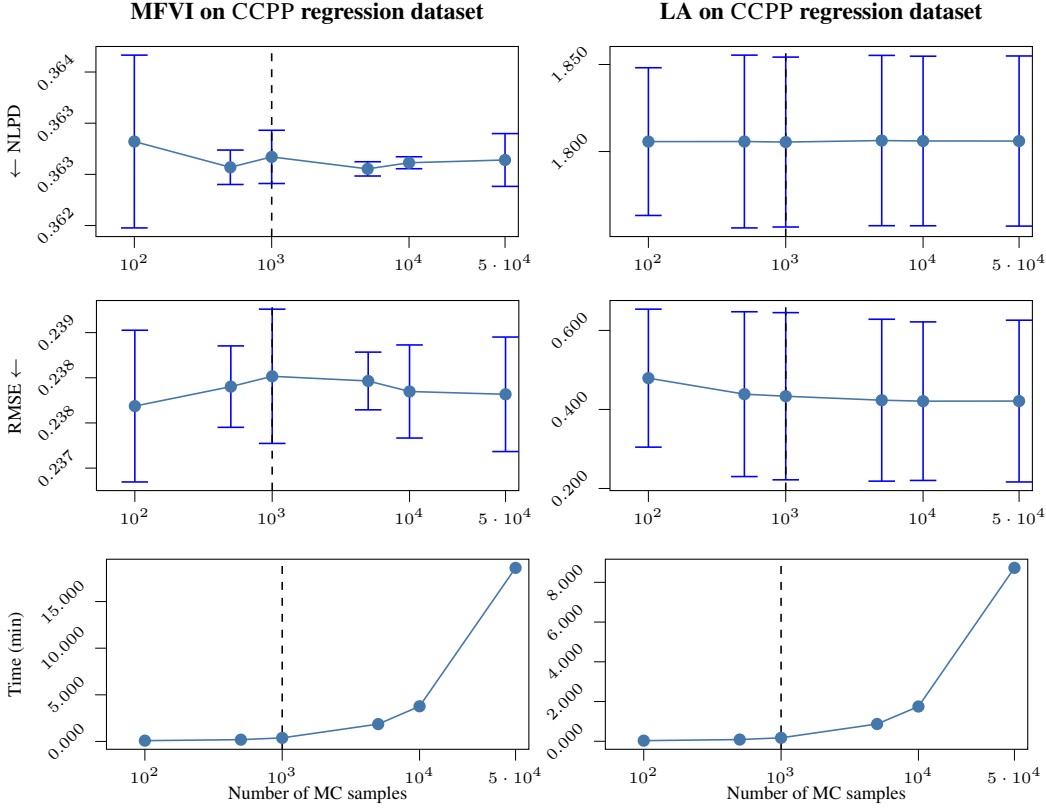

Figure 13: Effects of the number of MC samples on performance for LA and MFVI on regression tasks. In the results reported in main paper, we set the number of MC samples to 1000 (dashed line).

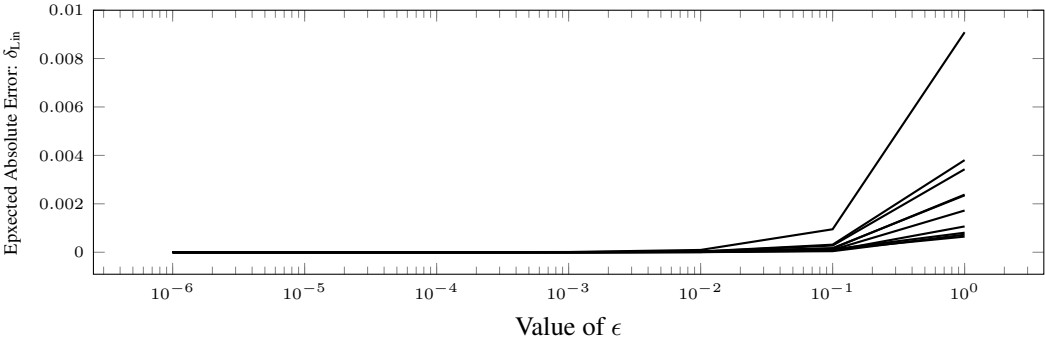

Figure 14: Estimated divergence from a locally linear function as a function of $\epsilon$. Note that a value of zero means that the function behaves locally like a linear function.

## B.7 RUNTIME EXPERIMENT

We compared the runtime of our method against sampling (using the 'torch-laplace' library Daxberger et al. (2021a) for Laplace and the IVON Shen et al. (2024)) and the GLM implementation of the 'torch-laplace' library for diagonal posterior covariances. For our streamlined approach on ViT, we assessed two cases: (i) propagating $\mathbb{C}\text{ov}[v_k, v_l]$ covariance terms (*cf.*, Eq. (74)) through the transformer (+Cov), and (ii) ignoring $\mathbb{C}\text{ov}[v_k, v_l]$ covariance terms. We used a pre-trained ViT base model on CIFAR-10 and a pre-trained MLP on MNIST. For comparison we also list the runtime for a single forward pass. For this, we ran experiments on an NVIDIA H100 80GB GPU for 400 data points, batchsize of one, and for each data point we repeated the measurement ten times. To account

Table 16: Expected Calibration Error (ECE) for ViT with posterior approximation n the attention layers. We compare only considering variance for value $V$ and considering full covariance. For MFVI and LA, both approximations results in almost the same result.

| Dataset | Mean Field Variational Inference | | Laplace Approximation | |
|---|---|---|---|---|
| | Full Covariance | Only Variance | Full Covariance | Only Variance |
| CIFAR-10 | 0.007 | 0.008 | 0.009 | 0.008 |
| CIFAR-100 | 0.024 | 0.026 | 0.027 | 0.023 |
| DTD | 0.040 | 0.040 | 0.042 | 0.043 |
| RESISC | 0.018 | 0.016 | 0.017 | 0.020 |
| IMAGENET-R | 0.043 | 0.132 | 0.130 | 0.039 |

for code compilation overheads, we dropped the first run on each data point. We report the mean and standard deviation of the runtime (in milliseconds) over the remaining nine runs and all $400$ data points. The results are shown in Table 17. For ViT, we can see that our method without $\mathbb{C}\mathrm{ov}[v_k, v_l]$ covariance terms has a comparable runtime to a single forward pass in the deterministic model. When additionally accounting for covariance terms, we obtain slight speed improvements over GLM but overall comparable performance. Note that our implementation is not optimised for speed and larger speedups may be obtained by optimising the code. For MLP, we obtain slight speed improvements over LA GLM but overall comparable performance.

Table 17: Wallclock times for ViT base on CIFAR-10 and MLP on MNIST in milliseconds.

| Model | Methods | AVG. RUNTIME ($\pm$ STD) $\downarrow$ |
|---|---|---|
| ViT | MAP | $3.737_{\pm 0.093}$ |
| | LA Sampling | $190.806_{\pm 0.137}$ |
| | LA GLM | $17.191_{\pm 0.734}$ |
| | MFVI Sampling | $207.854_{\pm 0.307}$ |
| | Ours (+ Cov) | $14.728_{\pm 0.144}$ |
| | Ours | $4.350_{\pm 0.079}$ |
| MLP | MAP | $0.069_{\pm 0.001}$ |
| | LA Sampling | $98.584_{\pm 3.737}$ |
| | LA GLM | $1.656_{\pm 0.049}$ |
| | MFVI Sampling | $190.302_{\pm 0.466}$ |
| | Ours | $0.542_{\pm 0.073}$ |

