# OpenReview forum: "Streamlining Prediction in Bayesian Deep Learning"
_ICLR.cc/2025/Conference — ICLR 2025 Poster_

### Official Review · Reviewer_tEnH · 2024-10-21

**Soundness:** 4
**Presentation:** 4
**Contribution:** 4
**Rating:** 8
**Confidence:** 4

**Summary:**

This paper proposes a simple way to obtain the a posteriori distribution over functions represented by Bayesian neural nets (BNNs). The core idea is to assume the product $W^{l} a^{l-1}$ between the weight matrix of layer $l$ and the activations $a^{l-1}$ from the previous layer to be a Gaussian. It follows that the preactivation $h^{l}$ is also Gaussian. Then, by linearizing the activation function $g$, one can approximate the activation vector $a^{l}$ is also Gaussian.

By induction, the output $a^{L} = f(x)$ is also Gaussian, which can readily be used to do inference in close form. For example, one can apply the probit approx. in classification or used in standard acquisition functions for sequential decision-making.

The key difference between this approach and previous approaches is the fact that the linearization is done in terms of the activations (and hence the input), and not over the weights as done in the linearized Laplace approximation. Nevertheless, the authors showed that the performance did not suffer from this linearization.

The important advantage of this approach is its scalability: It only requires a single forward pass with minimal overhead. This is in contrast to MC integration, which requires multiple forward passes, and the linearized Laplace, which requires a backward pass to compute the Jacobian over the weights. Indeed, the authors showed that their method is scalable to vision transformers and (small) language models.

**Strengths:**

- The proposed method is conceptually simple
- The proposed method might be (see weaknesses below) more efficient than prior methods in obtaining the output distribution.
- While linearization $x \mapsto f(x)$ is performed, the performance doesn't seem to suffer. In fact, it is very close to the linearized Laplace predictive (GLM in the paper).
- The method seems to be scalable (see weaknesses below).

**Weaknesses:**

I always appreciate the push for simple methods to make BNNs scalable without sacrificing performance. However, I noticed some issues in the presentation of the paper:

- The idea of linearizing the activation function has been done before in the context of continual learning by [Dhawan et al., ICML 2023](https://proceedings.mlr.press/v202/dhawan23a.html). The authors should cite this work and compare their method in detail. I acknowledge that the purposes are different. However, one cannot ignore the fact that they are very similar.

- It is unclear to me why one can still maintain good performance when the network is effectively linear in its inputs. Intuitively, this should not be the case since the representation learning of the base network is negated by the linearization. Could the authors please elaborate in detail (additional, empirical investigations like measuring the degree of linearity of the resulting linearized $f(x)$ might be one way to go about this).

- The main selling point of the approach is that to obtain the distribution over outputs, only a single pass is needed. However, it's currently unclear how much speedup this method provides compared to, e.g., the linearized Laplace (GLM).  A plot/table comparing the wall-clock times between the proposed method and the GLM predictive would be great.

- To make your selling point stronger, I suggest additional experiments with larger models such as Llama-3 7B etc. Laplace can still be done efficiently over PEFT (see [this](https://proceedings.mlr.press/v235/kristiadi24a.html) and [this](https://openreview.net/forum?id=FJiUyzOF1m&noteId=Uazck8TQ5A)). I believe this functionality is also available in `laplace-torch`: <https://aleximmer.github.io/Laplace/huggingface_example/>. Then, again, please compare the wall-clock times.

**Questions:**

Please see the Weaknesses section above. Additionally:

- How plug-and-play is the proposed method? Suppose the user has their model that has been Laplace-approximated via `laplace-torch`. What does the user need to do to enjoy the proposed method?

---

> ### Author Response · Authors · 2024-11-19
> **Rebuttal**
>
> We thank reviewer tEnH for their thorough review. We appreciate that they found our method conceptually simple. We address the comments and questions below.
>
> > **C1**: The idea of linearizing the activation function has been done before in the context of continual learning by Dhawan et al., ICML 2023. The authors should cite this work and compare their method in detail.
>
> Thank you for pointing this out. We agree *Dhawan et al.  (ICML 2023)* is a relevant related work and we have added discussion on it in the related work.
>
> > **C2**: It is unclear to me why one can still maintain good performance when the network is effectively linear in its inputs. Intuitively, this should not be the case since the representation learning of the base network is negated by the linearization. Could the authors please elaborate in detail (additional, empirical investigations like measuring the degree of linearity of the resulting linearized might be one way to go about this).
>
> In our approximation, the network is only locally linear and we do not globally linearise the network. Hence, we still capture interactions between the model parameters and globally capture the non-linear nature of the original model. This means we assume that the change of the function output behaves linearly locally around the input.
>
> Could the reviewer elaborate on the proposed: *"empirical investigations like measuring the degree of linearity of the resulting linearized"*?
> One way to locally assess the induced error would be in terms of an $L_p$ norm between the original and the approximate network in spirit of metrics used in certified robustness, e.g., [1].
>
>
>
> > **C3**: The main selling point of the approach is that to obtain the distribution over outputs, only a single pass is needed. However, it's currently unclear how much speedup this method provides compared to, e.g., the linearized Laplace (GLM)...
>
> We are currently conducting an empirical analysis of the efficiency of our method. We will post the results as soon as we have them and update the paper accordingly.
>
> > **C4**: To make your selling point stronger, I suggest additional experiments with larger models such as Llama-3 7B etc. [...]
>
> We agree that applying our method to larger language models like Llama-3 [2] is an exciting direction. While extending our work to, for example, Llama-3 is possible, it is due to the architectural choices a non-trivial task. In particular, the choice of the SwiGLU activation functions [3] $\operatorname{SwiGLU}(x, W, V, b, c, \beta)=\operatorname{Swish}_\beta(x W+b) \otimes(x V+c)$, where $W, V, b, c, \beta$, which is not a scalar activation function anymore and can lead to additional computational challenges. Similar challenges are to be expected w.r.t. the RMSNorm [4]. Propagating distributions through these parts may require additional approximations to obtain an efficient algorithm. Nevertheless, in the paper, we applied our method to GPT-2 and ViT as an important initial step to show that our method is indeed scalable to large-scale models (see Section 4.2). Extending our work to larger families of models, such as Llama-3, is an interesting future direction.
>
> > **Q1**: Suppose the user has their model that has been Laplace-approximated via laplace-torch. What does the user need to do to enjoy the proposed method?
>
> We plan to package our method into a library and integrate our method with the `laplace-torch` library for easy use after acceptance.
>
> ---
> References:
> [1] Changming Xu and Gagandeep Singh. "Cross-input Certified Training for Universal Perturbations", In ECCV 2024.
>
> [2] Touvron, Hugo, Thibaut Lavril, Gautier Izacard, Xavier Martinet, Marie-Anne Lachaux, Timothée Lacroix, Baptiste Rozière et al. "Llama: Open and efficient foundation language models." arXiv preprint arXiv:2302.13971 (2023).
>
> [3] Shazeer, Noam. "Glu variants improve transformer." arXiv preprint arXiv:2002.05202 (2020).
>
> [4] Zhang Biao and Rico Sennrich. "Root mean square layer normalization." In NeurIPS 2019.

---

> > ### Comment · Reviewer_tEnH · 2024-11-20
> >
> > Thanks for the reply! All good with most points, but I'd like to add:
> >
> > 1. Re. local linearization, even when you're not explicitly globally linearizing the network, doesn't $f(x)$ is still close to globally linear since the composition of linear functions is also linear? Because of this I'm interested in testing the linearity of the resulting approximation, e.g. by checking the finite-difference gradient $\partial f(x) / \partial x$, or by checking $f(x_1 + x_2) \approx f(x_1) + f(x_2)$ and $f(c x) \approx c f(x)$ for constant $c$. In the latter case, the "degree of linearity" would be $|f(x_1 + x_2) - f(x_1) - f(x_2)|$ and $|f(c x) - c f(x)|$, respectively. Happy to see other measures if you have better ideas!
> > 2. Looking forward to seeing the wall-clock results. I believe that these are crucial for the paper.
> > 3. Point taken on Llama. Though, you can try using this: <https://huggingface.co/SparseLLM/ReluLLaMA-7B>. It will make your work even stronger & impactful.

---

> > > ### Author Response · Authors · 2024-11-22
> > >
> > > > Local linearization, even when you're not explicitly globally linearizing the network, doesn't is still close to globally linear since the composition of linear functions is also linear?
> > >
> > > One way to understand the resulting locally linear function is as a piecewise linear function (or multi-linear function). Globally, the function will still be non-linear, but locally it will behave linearly. Also, note that piecewise linear functions are universal function approximators. In contrast to the original model, which composes piecewise linear functions if it is a ReLU network, our approximation composes linear functions *locally*. Hence, we still obtain a piecewise linear function as the linear function composition is performed locally and not globally. We added a paragraph providing an intuition on the approximation to the paper.
> > >
> > > > I'm interested in testing the linearity of the resulting approximation.
> > >
> > > Thank you for the suggested measures. We have added results for an experiment in which we assess the local linearity of a trained ReLU network in Appendix B.5.
> > >
> > > > Looking forward to seeing the wall-clock results. I believe that these are crucial for the paper.
> > >
> > > We have added wall-clock time results for an MLP and a ViT Base architecture to Appendix B.7. Our method obtains clear speedups compared to other approaches. We want to mention that our code is currently not optimised for speed and we expect that the runtimes will improve after packaging the code into a library.
> > >
> > > > Point taken on Llama. Though, you can try using this: https://huggingface.co/SparseLLM/ReluLLaMA-7B. It will make your work even stronger & impactful.
> > >
> > > Thank you for pointing this out, this looks indeed promising. We will have a look at it.

---

> > > > ### Comment · Reviewer_tEnH · 2024-11-22
> > > >
> > > > Thanks for the clarification and for the new results! I updated my score and support more strongly for an acceptance.
> > > >
> > > > Just additional comments:
> > > >
> > > > - For the intuition part, you can also use a two-layer NN with 1D input space for illustration. In the standard ReLU network case, one can think of the first layer as "arranging" ReLU functions on the input space (weight controls the steepness of the linear part of each ReLU along with horizontal reflection; bias moves the "kink" around). Then, the last layer's weight performs vertical reflection and linearly combines all ReLUs into a single piecewise linear function, and bias adjusts the final height of the resulting function. Your approximation also works similarly, intuitively speaking. Would be interesting to visually see the difference between your approximation and the original ReLU network in terms of features and the resulting piecewise-affine functions. See Fig. 1 of [1] for inspiration.
> > > >
> > > > - The speedups look really good. As a user, I'd be looking forward to seeing this approximation integrated with popular libraries such as `laplace-torch`!
> > > >
> > > > - I don't need to see the new results for ReLULlama-7B at this point, but your future readers will find it compelling when they see it. In conjunction with an integration into a popular Bayesian deep learning library, this will increase the adoption of your method.
> > > >
> > > > **References**
> > > >
> > > > [1] Kristiadi, Agustinus, Matthias Hein, and Philipp Hennig. "An infinite-feature extension for Bayesian ReLU nets that fixes their asymptotic overconfidence." NeurIPS 2021.

---

> > > > > ### Author Response · Authors · 2024-11-25
> > > > >
> > > > > Thank you for raising your score!
> > > > >
> > > > > > For the intuition part, you can also use a two-layer NN with 1D input space for illustration. [...]
> > > > >
> > > > > Thank you for the suggestion, we will include a similar figure into the paper.
> > > > >
> > > > > > I don't need to see the new results for ReLULlama-7B at this point, but your future readers will find it compelling when they see it. In conjunction with an integration into a popular Bayesian deep learning library, this will increase the adoption of your method.
> > > > >
> > > > > Thank you again for pointing us to ReLULlama.

---

### Official Review · Reviewer_oXsH · 2024-10-29

**Soundness:** 4
**Presentation:** 4
**Contribution:** 3
**Rating:** 8
**Confidence:** 4

**Summary:**

The paper proposes a method for computing an efficient approximation to the predictive distribution given a Bayesian neural network’s (BNN) posterior over the model’s weights. Their method locally linearises the activation functions, allowing for an analytical computation of the predictive distribution assuming a Gaussian weight space posterior approximation. This only requires a single forward pass, in contrast to MC sampling-based methods, and no expensive Jacobians that are necessary when globally linearising a neural network. The application of the method to different architectures and posterior covariance structures is discussed. To demonstrate the applicability of their method, the authors conduct extensive experiments applying their method to Laplace and mean-field variational inference posterior approximations on a range of architectures and datasets.

**Strengths:**

Overall, I recommend to accept the paper since it 1) proposes a simple, yet effective solution to the problem of efficient Bayesian neural network prediction, 2) appears to be technically correct and is well presented, and 3) provides extensive empirical evidence for the claim that their method performs at least comparable to more expensive alternatives.

The proposed method provides an intuitive solution to the problem of efficient BNN prediction, assuming that we want to consider a posterior distribution over more than just the last-layer weights of the neural network. The writing is clear, the figures and tables are nicely done. I also appreciate the detailed derivations in the appendix. While I have some concerns regarding the baselines, the experiments cover significant breadth. In particular, I like that that the sensitivity analysis was included, as this application highlights the more unique strengths of this method (see weaknesses for more on this).

**Weaknesses:**

The main weakness of the paper is that it does not consider alternative methods for approximating the predictive distribution of a BNN with a single forward pass. You do not compare the method to last-layer variants of the Laplace approximation and MFVI. This is important to determine if there is any benefit in trying to linearise the neural network to allow for more efficient predictions vs. just treating the network up to the last layer deterministically. For example, when using a last-layer Laplace approximation together with a probit approximation (in the case of classification), making a prediction also only requires a single forward pass and no expensive Jacobian computations. However, it is nice that you include results for which the linearisation w.r.t. the inputs is necessary, i.e. the sensitivity analysis. I think this could be even emphasised a bit more by explicitly saying that this is a downside of using last-layer methods.
Still, in most cases last-layer methods would be sufficient -- in particular also for the larger scale experiments in section 4.2.

Finally, I think the title is too general. As you describe yourself in the introduction, you can deconstruct Bayesian inference into three steps, 1) the specification of the prior, 2) the estimation of the posterior distribution, and 3) the computation of the predictive distribution. Your paper aims to streamline the third step. Hence, a more suitable title appears be “Streamlining prediction in Bayesian deep learning”.

**Questions:**

**Minor comments and questions**

- When mentioning stable distributions, please define them informally in one sentence first.
- Have you considered applying this method to architectures that use convolutions?
- I find the word “useful” in Figure 1 confusing. What makes this “useful” outlier detection/sensitivity analysis? What is non-useful outlier detection/sensitivity analysis?
- In the paragraph about ensemble methods in the related work section you mention that they typically require multiple forward passes. However, there are efficient ensembling approaches that only require a single forward pass and could be mentioned here, e.g. MIMO [1] and the combination of MIMO with last-layer Laplace approximations [2].
- One potential disadvantage of the method is the implementation overhead, as it requires a custom implementation for each layer-type within an architecture. Have you thought about how to improve the usability of your method?

[1] Havasi et al. (2021). [Training independent subnetworks for robust prediction](https://openreview.net/forum?id=OGg9XnKxFAH).

[2] Eschenhagen et al. (2021). [Mixtures of Laplace Approximations for Improved Post-Hoc Uncertainty in Deep Learning](https://arxiv.org/abs/2111.03577).

---

> ### Author Response · Authors · 2024-11-19
> **Rebuttal**
>
> We thank reviewer oXsH for their detailed and down-to-point review. We appreciate that they found our proposed method simple yet effective and that the experiments cover significant breadth and appreciate the sensitivity analysis. We address both comments and questions below.
>
> > **C1**: The main weakness of the paper is that it does not consider alternative methods for approximating the predictive distribution of a BNN with a single forward pass. You do not compare the method to last-layer variants of the Laplace approximation and MFVI. This is important to determine if there is any benefit in trying to linearise the neural network to allow for more efficient predictions vs. just treating the network up to the last layer deterministically.
>
> First, we want to mention that in the case of last layer Laplace, we recover the method by [1].
> Moreover, even though last layer Laplace approximation have empirically shown to be successful, they are not universal conditional density estimators [2], which might limit their effectiveness in practice. Hence, to assess the quality of our local linearisations we compared against moment-matching, as performed in DVI [3], with results shown in Table 3. We are happy to add additional results on last layer Laplace to the paper.
>
> > **C2**: I think the title is too general.
>
> We agree that our current title might not be specific enough to reflect the content, and we have changed the title as suggested.
>
> > **Q1**: When mentioning stable distributions, please define them informally in one sentence first.
>
> Thank you for pointing this out, we added the definition in the paper.
>
> > **Q2**: Have you considered applying this method to architectures that use convolutions?
>
> We considered a transformer architecture for vision tasks due to its popularity. Applying our method to convolution layers is an interesting avenue and we will add details into the appendix as soon as possible.
>
> > **Q3**: I find the word “useful” in Figure 1 confusing. What makes this “useful” outlier detection/sensitivity analysis? What is non-useful outlier detection/sensitivity analysis?
>
> We have changed the wording in Figure 1 to avoid unnecessary confusion.
>
> > **Q4**: In the paragraph about ensemble methods in the related work [...] there are efficient ensembling approaches that only require a single forward pass and could be mentioned here [...].
>
> Thank you for the suggested related works. We have added the suggested references on single forward-pass ensemble approaches to Section 2 in the revised paper.
>
> > **Q5**: Have you thought about how to improve the usability of your method?
>
> We will release the code publicly and plan to package our method into a library for easy use upon acceptance.
>
> ---
> References:
> [1] Agustinus Kristiadi, Matthias Hein, and Philipp Hennig. "Being Bayesian, even just a bit, fixes overconfidence in ReLU networks." In ICML 2020.
>
> [2] Mrinank Sharma, Sebastian Farquhar, Eric Nalisnick, and Tom Rainforth. "Do Bayesian Neural Networks Need To Be Fully Stochastic?" In AISTATS 2023.
>
> [3] Anqi Wu, Sebastian Nowozin, Edward Meeds, Richard E. Turner, José Miguel Hernández-Lobato, and Alexander L. Gaunt. "Deterministic variational inference for robust Bayesian neural networks". In ICLR 2019.

---

> > ### Comment · Reviewer_oXsH · 2024-11-20
> >
> > Thank you for addressing my concerns. I still think it would be valuable to compare to a last-layer Laplace approximation, especially in the larger scale experiments in section 4.2. Also, as mentioned by reviewer tEnH, an empirical analysis of the computational cost of your method vs. last-layer/GLM Laplace would be interesting here. I will keep my score as it is.
> >
> > One more suggestion: reviewer 8AJz pointed out that tuning the scaling factor requires validation data -- it might potentially be possible to use a Laplace approximation of the marginal likelihood to tune this hyperparameter without the need for validation data, at least when you use a Laplace approximation to fit the approximate posterior distribution over the weights [1][2].
> >
> > [1] Immer et al. (2021). [Scalable Marginal Likelihood Estimation for Model Selection in Deep Learning](https://arxiv.org/abs/2104.04975).
> >
> > [2] Daxberger et al. (2021). [Laplace Redux -- Effortless Bayesian Deep Learning](https://arxiv.org/abs/2106.14806).

---

> ### Author Response · Authors · 2024-11-22
>
> > I still think it would be valuable to compare to a last-layer Laplace approximation, especially in the larger scale experiments in section 4.2. [...]
>
> We are starting the comparison experiment for last-layer Laplace approximation for ViT Base now, and post the result as soon as possible.
>
> > an empirical analysis of the computational cost of your method vs. last-layer/GLM Laplace would be interesting here.
>
> We have added a wall-clock time results for a Bayesian MLP (c.f., Section 4.1) and a partially Bayesian ViT base (c.f., Section 4.2) to the Appendix B.7. In the case where more layers are treated Bayesian, we have comparable wall-clock with GLM and faster than sampling. We will add wall-clock times for last-layer Laplace after the experiment is finished.
>
> > reviewer 8AJz pointed out that tuning the scaling factor requires validation data -- it might potentially be possible to use a Laplace approximation of the marginal likelihood to tune this hyperparameter without the need for validation data, at least when you use a Laplace approximation to fit the approximate posterior distribution over the weights
>
> Thank you for the suggestion. It would indeed be interesting to explore the approximated marginal likelihood from LA to tune the scaling factor. We added details to the paper.
>
> > Have you considered applying this method to architectures that use convolutions?
>
> We have added a derivation for convolutional layers in Appendix A.7.

---

> > ### Comment · Reviewer_oXsH · 2024-11-22
> >
> > Thank you again for carefully addressing all my comments. I have increased my confidence to 4.

---

> > > ### Author Response · Authors · 2024-11-25
> > >
> > > Thank you for raising your confidence score.
> > >
> > > We have added the comparison results for the last layer Laplace approximation to Appendix B.2 (page 25). Compared to the case where more layers are treated Bayesian, last-layer approximations generally have lower accuracies and higher NLPD and ECE, which indicates the benefits gained by treating more layers Bayesian.

---

### Official Review · Reviewer_KRRC · 2024-10-31

**Soundness:** 3
**Presentation:** 3
**Contribution:** 1
**Rating:** 6
**Confidence:** 3

**Summary:**

The paper proposes an alternative to Monte Carlo sampling to compute the predictive distribution of a Bayesian neural network. This is obtained with a single forward pass instead of the multiple ones required by sampling. Some approximations make this possible, namely: Local linearization of activation functions, Gaussian approximation of linear layers, independence assumptions between activations and following layer parameters, deterministic key and query in attention layers and block diagonal structure of multi-head input.
On a variety of task, they empirically show superior or matching performances versus LA sampling, GLM and MFVI sampling.

**Strengths:**

The experiment sections cover a variety of tasks.

**Weaknesses:**

The main claim of the paper is to provide a scalable alternative to sampling, but they fail in providing sufficient evidence for such claim. Not all the experiments details are reported, first of all, they do NOT specify the number of samples used in the MC sampling baseline.
For whatever fixed parameter approximate posterior $q$, the proposed method is an approximation of the predictive distribution. Such predictive distribution is exactly what we get in the limit of infinite number of samples with MC sampling. Consequently, MC sampling has to perform better than the proposed method if enough samples are used.

This is point is not studied and not even addressed. Clearly indicating a malicious behaviour.
If the claim of the authors is that their method beats MC sampling "when not too many samples are used" then this would make sense, but this is neither the claim of the authors, neither such "too many" value is studied or attempted to show.

Another minor point is that in Line 230-231 the authors write "residual connection ... resulting distribution can be obtained analytically". Residual connection is essentially a sum, and the elements involved here are two Gaussians. While it is true that the sum of two Gaussian can be "obtained analytically", this step hide a big approximation: you can't store a mixture of Gaussians here, so the only scalable approach is to once again employ independence and "sum" the Gaussians erasing most of the information. While such approximation may be ok if it leads to good empirical performance, the way is presented is not clear and tends to hide such approximation step.

Typo and minor details:
-Line 80 "recent work (2024) developed ... and have shown good performance (2016) (2019)". I get what you mean, but the way the sentence is formulated imply that something developed in 2024 was used in 2016 and 2019...
-Line 260 "extends our the discussion" I guess this is a typo
-Line 324 "ours results in"  should be either "our results in" or "ours result in"

**Questions:**

Can the author explain why they never mention the number of samples used with MC sampling baseline?

Can the author provide an ablation study on number of samples used?

---

> ### Author Response · Authors · 2024-11-19
> **Rebuttal**
>
> We thank reviewer KRRC for the thoughtful comments and discussion and thank you for pointing out typos. We address the comments and questions below.
>
> > **C1**: "[...] they do NOT specify the number of samples used in the MC sampling baseline."
>
> We thank the reviewer for pointing out the missing detail about the number of samples used in the MC sampling baselines. This was not done intentionally. We have added the information on the number of MC samples for every experiment in Section 4. For both MFVI and LA Sampling, we used $1000$ samples in regression and classification experiments in Section 4.1, and $50$ samples for ViT and GPT models in Section 4.2.
>
>
> > **C2**: "Can the author provide an ablation study on number of samples used?"
>
> We have added an ablation study on the number of samples for the MC sampling baselines into Appendix B.4 of the revised paper. We show the ACC and NLPD evaluated on the test set, and measured the wall-clock time for evaluating the whole test set for different numbers of MC samples.
>
> Moreover, we want to point out that MC sampling has shown to be problematic in Laplace approximations, e.g., [1,2], and results in a computational hurdle for larger models. Our work focuses on making predictions in Bayesian deep learning more practical by providing a computationally feasible approximation to the posterior predictive.
>
>
> > **C3**: "Residual connection is essentially a sum, and the elements involved here are two Gaussians [...] this step hide a big approximation: you can't store a mixture of Gaussians here [...]"
>
> We believe that there might be a misunderstanding. In the case of linear layers and residual connections, we end up with a sum of Gaussian random vectors (RVs) rather than their distribution. For residual connections, this consists of Gaussian RVs representing the inputs and outputs of a non-linear layer due to the skip connection. To obtain the distribution over the outputs of the residual connection, we assume independence between the input RV and output RV of the non-linear layer. Hence, the output RV follows a Gaussian distribution. We agree that assuming independence is a strong assumption, especially for the residual blocks, but find that this empirically works well while remaining computationally feasible. We elaborate on this in the limitations in Section 5, and a detailed derivation for the case of a linear layer is given in Appendix A.1. Please let us know if there are any further questions.
>
> > Line 80 "recent work (2024) developed ... and have shown good performance (2016) (2019)".  I get what you mean, but the way the sentence is formulated imply that something developed in 2024 was used in 2016 and 2019
>
>
>
> To clarify, the work by [3] (2024) is the first time MFVI was successfully applied to the large scale models [4] (2016) and [5] (2019). We believe there might have been a misunderstanding. For completeness, below we are quoting line 80 from our submission:
> >> Recent work by Shen et al. (2024) developed an optimiser to ease the use of MFVI, and have shown good performance on large-scale models such as ResNets (He et al., 2016) and GPT (Radford et al., 2019).
>
> ---
> References:
> [1] Lawrence, Neil David. "Variational inference in probabilistic models." PhD diss., University of Cambridge, 2001.
>
> [2] Alexander Immer, Maciej Korzepa, and Matthias Bauer. "Improving predictions of Bayesian neural nets via local linearization". In AISTATS 2021.
>
> [3] Yuesong Shen, Nico Daheim, Bai Cong, et al. "Variational Learning is Effective for Large Deep Networks". In ICML 2024.
>
> [4] Saining Xie, Ross Girshick, Piotr Dollár, Zhuowen Tu, and Kaiming He. "Aggregated Residual Transformations for Deep Neural Networks". In CVPR 2016.
>
> [5] Alec Radford, Jeffrey Wu, Rewon Child, David Luan, Dario Amodei, and Ilya Sutskever. "Language Models are Unsupervised Multitask Learners". OpenAI blog 2019.

---

> > ### Comment · Reviewer_KRRC · 2024-11-20
> > **Thanks! Increasing my score**
> >
> > I thank the authors for providing details on the number of samples used and also for adding the ablation studies I asked for. This rules out the malicious behaviour I was suspecting of, and completely solve my main concern. Well done! Based on this I'm increasing my score.
> >
> > Nonetheless, the concern about the independence assumption remains. With "sum of gaussians" I was referring to "the distribution of sum of random variable", which is what's going on in the residual connection. The independence assumption between input and output clearly does not hold for residual connections and thus I emphasize, again, that yes "assuming independence is a STRONG assumption". I agree that empirical evidence supports your method anyway, but this theory limitation should be highlighted more clearly, in my opinion.

---

> > > ### Author Response · Authors · 2024-11-20
> > >
> > > Thank you for the fast response and for increasing the score.
> > >
> > > Regarding the independence assumption made in residual connections, we have added a discussion about this limitation to the paper and will further improve the discussion after the additional experiments have finished.

---

### Official Review · Reviewer_8AJz · 2024-11-04

**Soundness:** 3
**Presentation:** 3
**Contribution:** 3
**Rating:** 6
**Confidence:** 4

**Summary:**

This paper aims to address the practical challenge of efficient uncertainty estimation in Bayesian deep learning (BDL). While recent work has improved posterior estimation in BDL, computing predictions typically requires expensive Monte Carlo sampling.
The authors propose a streamlined approach that enables analytical computation of predictions in a single forward pass. Their method combines Local linearisation of activation functions using first-order Taylor expansions and local Gaussian approximations at linear layers.

**Strengths:**

The paper introduces a novel approach to Bayesian Deep Learning (BDL) by eliminating the need for Monte Carlo sampling through local approximations. It nicely blends local linearisation with Gaussian methods and offers interesting solutions for managing transformer architectures and Kronecker-factored covariance.

Quality-wise, the work is solidly validated with extensive experiments across various tasks and architectures. It thoroughly benchmarks against established baselines, clearly showcasing benefits like improved efficiency and reliable uncertainty estimates. Additionally, the method is versatile, supporting different posterior approximations such as Laplace Approximations and MFVI.

The presentation is clear and well-organised.

**Weaknesses:**

The treatment of attention layers (using deterministic queries/keys) needs stronger justification. The scaling factor for predictive variance is tuned on validation data - this seems ad hoc. No discussion of potential failure modes or limitations of the local linearisation assumption

**Questions:**

Please see weaknesses above

---

> ### Author Response · Authors · 2024-11-19
> **Rebuttal**
>
> We thank reviewer 8AJz for their constructive review and appreciate they find our work novel, versatile, and solidly validated. We address both comments and questions below.
>
> > **C1**: The treatment of attention layers (using deterministic queries/keys) needs stronger justification.
>
> Recall that the attention score is computed as $\text{Softmax}(\mathbf{Q}\mathbf{K}^{\top})$. In case we treat queries and keys as random vectors, the softmax function will "squish" the uncertainties over queries and keys and not result in a random vector following a distribution close to a Gaussian. Henceforth, we decided to treat queries and keys deterministically, which also elevates a potential computational bottleneck. A possible remedy is to leverage an approximation to the softmax function such as [1]. We have added a discussion on this to Section 3.2.
>
>
> > **C2**: The scaling factor for predictive variance is tuned on validation data - this seems ad hoc.
>
> The use of our scaling factor is similar in notion to the pseudo-count used in [2]. But we agree that it is somewhat ad hoc. Hence, we mention the scaling factor in the Limitations paragraph in Section 5.
>
> > **C3**: No discussion of potential failure modes or limitations of the local linearisation assumption.
>
> We have added a Limitations paragraph in Section 5 of the revised paper, where we mention the induced error from local linearisation and independence assumptions, as well as the need for a validation set for fitting the scaling factor.
>
>
> ---
> References:
> [1] Jiachen Lu, Jinghan Yao, Junge Zhang, Xiatian Zhu, Hang Xu, Weiguo Gao, Chunjing Xu, Tao Xiang, Li Zhang. "SOFT: Softmax-free Transformer with Linear Complexity". In NeurIPS 2021.
>
> [2] Hippolyt Ritter, Aleksandar Botev, and David Barber. "A scalable laplace approximation for neural networks." In ICLR 2018.

---

> > ### Comment · Reviewer_8AJz · 2024-11-27
> > **Response**
> >
> > Thanks for your response, although I'll keep my score because of the discussed limitations.

---

### Meta-Review · Area_Chair_XPzc · 2024-12-20

**Metareview:**

This paper addresses computational challenges in Bayesian deep learning by proposing a method for approximating predictive distributions using a single forward pass. The method's effectiveness is demonstrated on a variety of architectures (MLPs, ViT, GPT-2) and tasks (regression, classification). Empirical results show the method achieves comparable or superior accuracy and uncertainty estimates relative to Monte Carlo sampling, with significant computational efficiency.

**Additional Comments On Reviewer Discussion:**

Reviewers gave positive feedback on the method's scalability and strong empirical validation across tasks and architectures. Concerns raised included missing details on Monte Carlo sampling baselines, the independence assumption for residual connections, and the lack of comparison to last-layer Laplace approximations. The authors effectively addressed these in their revision. Suggestions for further improvements included experimenting on larger models (e.g., ReLULlama-7B). All reviewers agree on the paper’s acceptance.

---

### Decision · Program_Chairs · 2025-01-22

Accept (Poster)